# DFVEdit: Conditional Delta Flow Vector for Zero-shot Video Editing

## Abstract

The advent of Video Diffusion Transformers (Video DiTs) marks a milestone in video generation. However, directly applying existing video editing methods to Video DiTs often incurs substantial computational overhead, due to resource-intensive attention modification or finetuning. To alleviate this problem, we present DFVEdit , an efficient zero-shot video editing method tailored for Video DiTs. DFVEdit eliminates the need for both attention modification and fine-tuning by directly operating on clean latents via flow transformation. To be more specific, we observe that editing and sampling can be unified under the continuous flow perspective. Building upon this foundation, we propose the Conditional Delta Flow Vector (CDFV) – a theoretically unbiased estimation of DFV – and integrate Implicit Cross Attention (ICA) guidance as well as Embedding Reinforcement (ER) to further enhance editing quality. DFVEdit excels in practical efficiency, offering at least 20x inference speed-up and 85% memory reduction on Video DiTs compared to attention-engineering-based editing methods. Extensive quantitative and qualitative experiments demonstrate that DFVEdit can be seamlessly applied to popular Video DiTs (*e.g.*, CogVideoX and Wan2.1), attaining state-of-the-art performance on structural fidelity, spatial-temporal consistency, and editing quality.

## 1 Introduction

In the wave of digitization, video creation has become a dominant form of entertainment. In response, research on controllable video generation holds considerable practical importance. While Video Diffusion Transformer (DiT) models Yang et al. (2024b); Kong et al. (2024); Wang et al. (2025) have revolutionized video synthesis quality, and DiT-based image editing methods Feng et al. (2024); Kulikov et al. (2024); Zhu et al. (2025); Dalva et al. (2024); Rout et al. (2024); Jiao et al. (2025) have achieved remarkable success, video editing remains challenging in preserving spatiotemporal fidelity. Critically, existing video editing methods do not fully exploit the capabilities of Video DiTs, limiting the potential for high-quality controllable video generation.

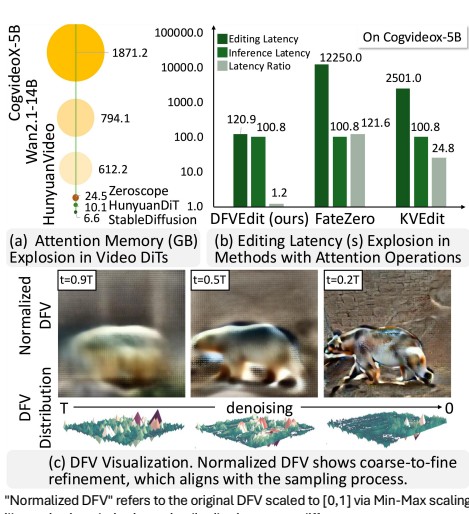

(a) Attention Memory (GB) Explosion in Video DiTs

(b) Editing Latency (s) Explosion in Methods with Attention Operations

(c) DFV Visualization. Normalized DFV shows coarse-to-fine refinement, which aligns with the sampling process.

"Normalized DFV" refers to the original DFV scaled to [0,1] via Min-Max scaling, illustrating its relative intensity distribution across different steps t.

Figure 1: Key insight and motivation.

Existing video editing techniques mainly follow two paradigms: training-based methods Singer et al. (2022); Wu et al. (2023); Shin et al. (2024); Liu et al. (2024c) and zero-shot methods Qi et al. (2023); Cai et al. (2024); Geyer et al. (2023); Zhang et al. (2023b); Yang et al.; Wang et al. (2024b). We focus on zero-shot methods for the plug-and-play flexibility and application efficiency. For training-free video editing, a high-quality pre-trained base model is crucial. Early video editing methods primarily utilized image diffusion models Rombach et al. (2022); Song et al. (2020a), which suffered from temporal inconsistencies due to the lack of capable video diffusion models. These early methods Khachatryan et al. (2023); Qi et al. (2023); Geyer et al. (2023); Yatim et al. (2024) not only had to ensure structural integrity and editing accuracy but also required significant effort to enhance temporal coherence. In contrast, methods Cai et al.

(2024); Ku et al. (2024) based on video diffusion models naturally excel in temporal consistency, leading us to leverage the latest Video DiTs Wang et al. (2025); Yang et al. (2024b); Kong et al. (2024) for video editing. Regardless of the type of base models, achieving high fidelity and temporal consistency hinges on attention engineering in most existing methods, including various attention caching and modification techniques. The key to effective attention engineering is that attentions (including keys, queries, and values) contain the spatial-temporal information of the source video, allowing for smooth editing of target regions while preserving the original content's integrity. However, attention mechanisms now consume hundreds of gigabytes of memory (Fig. 1(a)) in Video DiTs Yang et al. (2024b); Wang et al. (2025); Kong et al. (2024), a significant increase from previous usage in Unet-based diffusion models Wang et al. (2023); Rombach et al. (2022); Song et al. (2020a) and image DiT models Li et al. (2024b); Yang et al. (2024a) at the gigabyte scale. This suggests that traditional attention engineering techniques are incompatible with Video DiTs, creating an urgent need for methods that preserve editing quality while improving computational efficiency.

Motivated by this inefficiency, we shift the focus from attention to input latents and introduce a continuous flow transformation framework, DFVEdit, for direct video latent refinement. We observe that the standard sampling process in video diffusion models—whether based on Score Matching Song et al. (2020b) or Flow Matching Lipman et al. (2022)—can be unified under a continuous flow perspective. Based on this insight, we demonstrate that editing from the source to the target video naturally forms a time-dependent flow vector field (Fig. 1(c)), which we term the Delta Flow Vector (DFV).

Building upon this foundation, we introduce the Conditional Delta Flow Vector (CDFV) to estimate the flow from source to target latent, incorporating Implicit Cross Attention Guidance (ICA) and Embedding Reinforcement (ER) to further improve editing accuracy. The CDFV in Video DiTs inherently enforces spatial-temporal dependencies while its divergence directly determines update weights. This physically grounded formulation provides two fundamental advantages over approximation-based latent-refinement approaches like DDS Hertz et al. (2023) and SDS Poole et al.: (1) *theoretical unification* by modeling both sampling and editing from the continuous flow perspective and (2) *computational efficiency* through divergence-determined and hyperparameter-free weights that eliminate heuristic scheduling and overcome low convergence issues inherent to shallow approximations. Moreover, for the seamless application to video editing, we enhanced spatiotemporal coherence by intrinsically avoiding randomness bias while incorporating ICA guidance and ER mechanisms (Fig. 5). Experiments show DFVEdit achieves at least 20× speed-up and 85% memory reduction over attention-engineering-based methods on Video DiTs (*e.g.*, CogVideoX, Wan2.1), while maintaining SOTA performance in fidelity, temporal consistency, and editing quality. Consequently, our approach offers an efficient and versatile solution for zero-shot video editing on Video DiTs.

## 2 RELATED WORK

**Video Diffusion Transformer.** Video Diffusion Transformers have evolved from early 3D-UNet-based designs Zhang et al. (2023a); Wang et al. (2023); Blattmann et al. (2023); Chen et al. (2024) to modern 3D-Transformer-based designs Peebles & Xie (2023). Advanced models such as Open-Sora Zheng et al. (2024); Lin et al. (2024), CogVideoX Yang et al. (2024b), HunyuanVideo Kong et al. (2024) and Wan Wang et al. (2025) have all or part of the following key innovations: replacement of 3D-UNets with scalable 3D-Transformer blocks; integration of cross-attention and self-attention into a unified 3D-full-attention Yang et al. (2024b); Kong et al. (2024); and adoption of 3D-VAE Yang et al. (2024b) for spatiotemporal latent compression. Some Video DiTs Li et al. (2024b); Wang et al. (2025) are combined with Flow Matching Lipman et al. (2022) while others Yang et al. (2024b) adopt SDE Song et al. (2020b) samplers like DPM-solver Lu et al. (2022).

**Image editing on Diffusion Transformer.** With the rise of Diffusion Transformer Peebles & Xie (2023), DiT-based image editing methods Yang et al. (2024a); Li et al. (2024b) have emerged. However, directly applying image editing methods to videos often fails to address temporal consistency and motion fidelity. Additionally, adapting them to Video DiTs introduces extra challenges. Firstly, generalization limitations occur when applying methods Dalva et al. (2024); Kulikov et al. (2024); Rout et al. (2024); Jiao et al. (2025); Garibi et al. (2024); Deutch et al. (2024) that rely on rectified flow Esser et al. (2024) or distilled few-step models Sauer et al. (2024) to Video DiTs that are not combined with rectified flow or distillation techniques. Secondly, efficiency limitations are present for image editing methods Nguyen et al. (2024) that require finetuning. Furthermore, even generalized and efficient methods like DiT4Edit Feng et al. (2024) and KVEdit Zhu et al. (2025), which use

attention or key-value caching and modification, still face prohibitive computational costs due to the more massive attention overhead in Video DiTs compared to image DiTs.

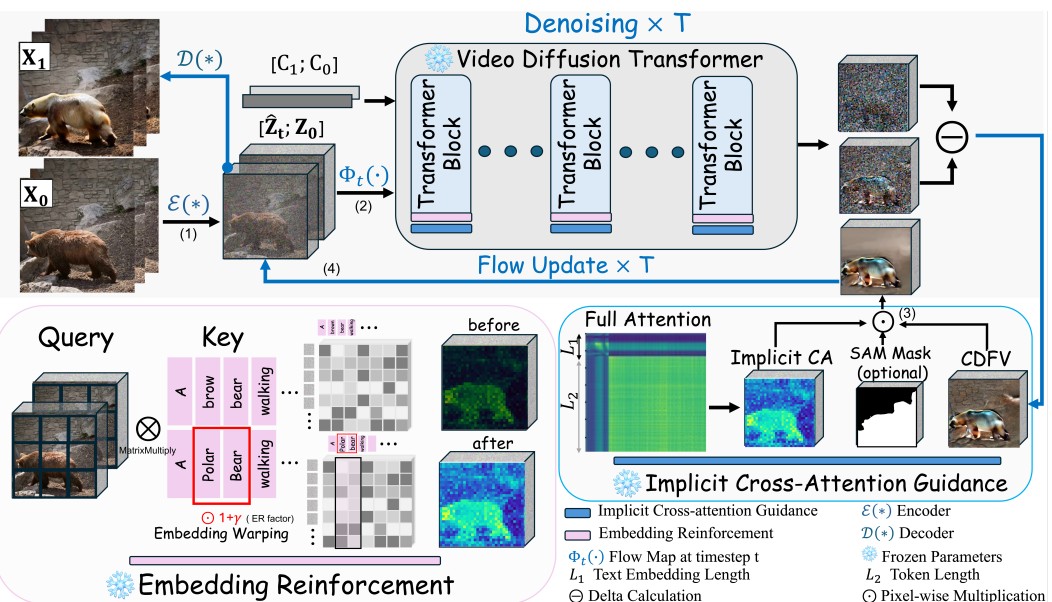

Figure 2: **DFVEdit overview.** Follow these steps for DFVEdit: (1) Encode $\mathbf{X}_0$ into the latent space $\mathbf{Z}_0$, and initialize the target latent variable as $\hat{\mathbf{Z}}_T = \mathbf{Z}_0$. (2) Transform $[\hat{\mathbf{Z}}_T; \mathbf{Z}_0]$ via the flow map $\Phi_T(\cdot)$. (3) Feed the result with prompt embeddings $[C_1, C_0]$ into the Video Diffusion Transformer, compute the delta difference to obtain the CDFV at timestep $T$, then refine it using ER and ICA. (4) Update $\hat{\mathbf{Z}}_T \rightarrow \hat{\mathbf{Z}}_{T-1}$ using the enhanced CDFV, and iterate (1)-(4) until reaching $\hat{\mathbf{Z}}_0$. (5) Decode $\hat{\mathbf{Z}}_0$ to generate the target video $\mathbf{X}_1$.

**Video editing.** Video editing via diffusion models is dominated by two paradigms: training-based and training-free methods. Training-based approaches Jiang et al. (2025); Peruzzo et al. (2024); Esser et al. (2023); Gu et al. (2024); Zi et al. (2025); Wang et al. (2024a); Wu et al. (2023); Liu et al. (2024c) enhance pre-trained image diffusion models Rombach et al. (2022) with spatiotemporal modules, optimizing for complex edits but at high computational costs, limiting real-time applications. Conversely, training-free methods emphasize computational efficiency and real-time capability. Training-free video editing commonly involves two stages: latent space initialization and editing condition injection. Latent space initialization typically follows three paradigms: (1) forward diffusion with some steps for preserving low-frequency features Meng et al. (2021); Yang et al. (2023), (2) DDIM Song et al. (2020a) inversion for enabling deterministic reconstruction Qi et al. (2023); Geyer et al. (2023), or (3) direct source latent usage Hertz et al. (2023); Poole et al.. For editing condition injection, most existing zero-shot methods heavily rely on *attention engineering* to maintain spatial-temporal fidelity. For instance, FateZero Qi et al. (2023) enhances temporal consistency by caching attention maps from DDIM Song et al. (2020a) inversion and integrating them into the denoising process; TokenFlow Geyer et al. (2023) improves spatiotemporal coherence by leveraging cached attention outputs from DDIM inversion for inter-frame correspondences and incorporating extended attention blocks during denoising; VideoDirector Wang et al. (2024b) achieves fine-grained editing via SAM Kirillov et al. (2023) masks by fusing self-attention with reconstruction attention and mask guidance; and VideoGrain Yang et al. realizes complex semantic structure modifications through SAM masks while operating on complex attention map modifications. These attention-engineered methods face scalability challenges in Transformer blocks Vaswani et al. (2017), particularly for Video DiTs Kong et al. (2024); Wang et al. (2025) where attention memory demands grow dramatically (Fig. 1). Moreover, approaches Yoon et al. (2024b); Liu et al. (2024a); Ku et al.; Yatim et al. (2024) free of attention engineering suffer from structural degradation: FRAG Yoon et al. (2024b) mitigates blurring and flickering through frequency processing but compromises fidelity due to basic DDIM inversion Song et al. for source content retention; DMT Yatim et al. (2024) employs SSM Yatim et al. (2024) loss for motion transfer yet underperforms in detail preservation; and first-frame propagation

methods (*e.g.*, StableV2V Liu et al. (2024a), AnyV2V Ku et al.) introduce accumulating artifacts without full-frame coordination. In conclusion, designing efficient and high-quality editing methods tailored for Video DiTs remains a critical challenge.

# 3 METHOD

Fig. 2 provides an overview of DFVEdit. Given a source video $\mathbf{X_0} \in \mathbb{R}^{F \times 3 \times H \times W}$ comprising $F$ RGB frames at resolution $H \times W$, together with source and target text prompts $(P_0, P_1)$, our method supports both global stylization and local modifications (shape and attribute editing). The edited video $\mathbf{X_1}$ preserves spatiotemporal integrity in unedited regions while ensuring motion fidelity and precise alignment with $P_1$. Our approach leverages two key insights: manipulating latent space is more computationally efficient than manipulating attention (Fig. 1), and editing can be modeled as the continuous flow transformation between the source and target videos (Sec 3.1). We introduce the Conditional Delta Flow Vector (CDFV) (Sec 3.2) for this transformation. To enhance video editing performance, we utilize Implicit Cross-Attention Guidance and Embedding Enforcement (Sec 3.3) to improve spatiotemporal fidelity.

## 3.1 UNIFIED CONTINUOUS FLOW PERSPECTIVE ON SAMPLING AND EDITING

Diffusion models include inverse and forward processes. The inverse process is typically parameterized as a Markov chain with learned Gaussian transitions, mapping noisy inputs to clean outputs. Conversely, the forward process gradually adds Gaussian noise to the clean input according to a variance schedule. As mentioned in Song & Ermon (2020; 2019); Song et al. (2020b), given a data input $x$, both inverse and forward processes can be regarded as overdamped Langevin Dynamics uhl (1930) (named Stochastic Differential Equation (SDE) in Score Matching Song et al. (2020b)):

$$dx_t = f(x_t, t)dt + g(x_t, t)dW \quad (1)$$

where $f(x_t, t)$ is the drift coefficient corresponds to deterministic direction and $g(x_t, t)$ is the diffusion coefficient corresponds to disturbing intensity and $dW$ is a Wiener process and the probability density $P(x_t, t)$ can be described by introducing the Fokker-Planck equation Jordan et al. (1998) combined with the Ito's lemma Kloeden et al. (1992) and the concept of probability flow:

$$\frac{\partial P(x_t, t)}{\partial t} = -\nabla \left[ \left( f(x_t, t) - \frac{g^2(x_t, t)}{2} \nabla log P(x_t, t) \right) P(x_t, t) \right] \quad (2)$$

Eq. 2 generalizes traditional sampling methods like DDPM Ho et al. (2020) and DDIM Song et al. (2020a). ***This formulation reveals that methods based on SDE Song et al. (2020b) obey the continuity equation principle of Flow Matching Lipman et al. (2022) and can be unified under a continuous flow perspective.*** The continuous flow is characterized by a vector field $v_t(x_t) = f(x_t, t) - \frac{g^2(x_t, t)}{2} \nabla log P(x_t, t)$, enabling state transitions from $x_t$ to $x_{t+\Delta t}$ either through flow map $\Phi_t$ in Eq. 3 or through its Euler discretized approximation in Eq. 4:

$$\begin{cases} \dfrac{d}{dt} \Phi_t(x) = v_t(\Phi_t(x)) \\ \Phi_0(x) = x \end{cases} \quad (3)$$

$$x_{t+\Delta t} = x_t + \Delta t * v_t(\Phi_t(x)) \quad (4)$$

As discussed in Section 2, zero-shot video editing includes two stages: latent space initialization and editing condition injection. The first stage involves a standard sampling process. In the second stage, we derive an isomorphism with sampling process by formulating video editing as:

$$X_{t-1}^{\text{edit}} = g_{\theta_{2,t}} \Big( X_t^{\text{edit}}, \underbrace{\epsilon_{\theta_1}(X_t^{\text{edit}}, t)}_{\text{Canonical Denoiser}} + \lambda \underbrace{C(X_t^{\text{edit}}, t, *)}_{\text{Control Term}} \Big) \quad (5)$$

where $\{X_t^{\text{edit}}\}_{t=0}^T$ defines the state trajectory of the edited video in the sampling process; $g_{\theta_{2,t}}$ is differentiable transition function parameterized by learnable $\theta_2$; $\epsilon_{\theta_1}$ is pretrained diffusion model with frozen $\theta_1$; $C(x, t, *)$ is the control term with intensity $\lambda \geq 0$ and optional extra input $*$. Under the Euler discretization scheme with step size $\Delta t \to 0$ and $\theta_2 = \mathcal{I}$, the discrete process in Eq. 17 converges to the controlled SDE:

$$dX_t^{\text{edit}} = \underbrace{\left[ -\frac{\beta(t)}{2} X_t^{\text{edit}} + \frac{\beta(t)}{2} \nabla \log p_t(X_t^{\text{edit}}) + \lambda \frac{\beta(t)}{2} \sigma(t) C(X_t^{\text{edit}}, t, *) \right]}_{f_{\theta_1}(X_t^{\text{edit}}, t)} dt + \underbrace{\sqrt{\beta(t)}}_{g(t)} dW \quad (6)$$

where $\nabla \log p_t(X_t^{\text{edit}})$ is the score function, and $\sigma(t) = \sqrt{(1 - \alpha(t))/\alpha(t)}$ is the signal-to-noise ratio coefficient with $\alpha(t) = e^{-\int_0^t \beta(s)ds}$. ***The structural isomorphism between Eq. 31 and the stochastic differential equation in Eq. 1 indicates that video editing processes can be represented within a continuous flow sampling framework***, as shown in Eq. 3 (see Appendix for more details).

## 3.2 Conditional Delta Flow Vector

Building upon the isomorphic correspondence between editing and sampling, we introduce the Conditional Delta Flow Vector (CDFV) to establish a direct continuous flow bridge from the source video to the target video.

**Delta Flow Vector.** Given the initial distribution $p(Z_T) = \mathcal{N}(Z_T; 0, I)$ for the reverse process and a clean video latent $Z$, Eq. 3 implies the existence of a time-dependent flow map $\Phi$ that:

$$Z = Z_T - \sum_{t=0}^{T} \Delta t v_t(\Phi_t(Z)) \tag{7}$$

Assuming the source and target latents $(\hat{Z}_0, Z_0)$ and their corresponding prompts $(P_1, P_0)$ are given, we replace $Z$ in Eq. 7 with $Z_0$ and $\hat{Z}_0$ respectively and define the Delta Flow Vector (DFV) as $\Delta v_t(\hat{Z}_0, Z_0) = v_t(\Phi_t(\hat{Z}_0)) - v_t(\Phi_t(Z_0))$, and the target latent $\hat{Z}_0$ can be expressed in terms of the source latent $Z_0$ as:

$$\hat{Z}_0 = Z_0 - \sum_{t=0}^{T} \Delta t \, \Delta v_t(\hat{Z}_0, Z_0). \tag{8}$$

***Eq. 8 establishes a continuous flow directly from the source latent $Z_0$ to the target latent $\hat{Z}_0$***, with the vector field defined as $v_t = \Delta v_t(\hat{Z}_0, Z_0)$. While prior works Han et al. (2024); Couairon et al. (2022); Hertz et al. (2023) heuristically observed that latent differences indicate editing regions, we rigorously prove this as a special case of DFV when the transformation state and vector field satisfy the continuity equation (Eq. 3).

**Conditional Delta Flow Vector.** The direct computation of $\Delta v_t(\hat{Z}_0, Z_0)$ is intractable since $\hat{Z}_0$ is the editing target. To resolve this problem, ***we leverage the terminal condition of diffusion processes to derive an unbiased estimation of DFV.*** From Eq. 2 we obtain $v_t(x_t) = f(x_t, t) - \frac{g^2(x_t,t)}{2}\nabla log P(x_t, t)$. As $t$ approaches $T$, and given that $P(x_t, t)$ is the probability density of $x_t$, if we set winner process of $Z_0$ and $\hat{Z}_0$ is equal, then $g(Z_0, t) = g(\hat{Z}_0, t)$. Consequently, as $t \to T$, both $P(Z_0, t)$ and $P(\hat{Z}_0, t)$ follow a normal distribution $\mathcal{N}(Z_T; 0, I)$ with zero mean and unit variance. Moreover, $\hat{Z}_t$ is equivalent to $Z_t$ as $t \to T$, and we have:

$$\Delta v_t(\hat{Z}_0, Z_0) \underset{t \to T}{=} f_{\theta_1, c_1}(Z_t, t) - f_{\theta_1, c_0}(Z_t, t) \tag{9}$$

The latent $\hat{Z}_{T-\Delta t}$ can be updated using Eq. 10, which corresponds to applying the continuous flow map from $\hat{Z}_0$ as defined in Eq. 11:

$$\hat{Z}_{T-\Delta t} = Z_{T-\Delta t} - \Delta t \left[ f_{\theta, c_1}(Z_T, t) - f_{\theta, c_0}(Z_T, t) \right], \tag{10}$$

$$\hat{Z}_{T-\Delta t} = \Phi_{T-\Delta t}(\hat{Z}_0). \tag{11}$$

We sequentially obtain all $v_t(\Phi(\hat{Z}_0))$ and define the Conditional Delta Flow Vector (CDFV) in Eq. 12.

$$\begin{cases} \Delta v_t(Z_0, c_0, c_1) = v_{t,c_1}(\hat{Z}_t) - v_{t,c_0}(\Phi_t(Z_0)) \\ \hat{Z}_T = \Phi_T(Z_0) \end{cases} \tag{12}$$

Theoretically, the CDFV provides an unbiased estimate of DFV. By using the CDFV as a control term, defined in Eq. 13, and integrating it into Eq. 31, we maintain a computational complexity similar to that of the basic sampling process. See the Appendix B for more details.

$$C(\hat{Z}_t, t, *) = \frac{\nabla \log P(\hat{Z}_t, t) - \nabla \log P(\Phi_t(Z_0), t)}{\sigma(t)} \tag{13}$$

### 3.3 SPATIOTEMPORAL ENHANCEMENT FOR CDFV

**Implicit Cross-Attention Guidance**. Although CDFV extracted from Video DiTs theoretically captures semantic differences between $P_0$ and $P_1$ with temporal coherence (Sec 3.2), empirical studies reveal persistent background leakage (Fig. 2). We attribute this phenomenon to the score function $\nabla_X \log p_t(X; \theta)$, which is learned by the model and may not perfectly align with theoretical expectations. This discrepancy can introduce local distributional drift in unedited regions, and such shifts have the potential to cause noticeable alterations in the background of edited videos (see Fig. 5 for examples). Segmentation masks play a crucial role in effective structure guidance, and cross-attention, as highlighted in Cai et al. (2024); Qi et al. (2023); Hertz et al. (2022), exhibit significant potential for shape editing tasks. This is attributed to their time-aware adaptability and target-following characteristics, which enhance the capability to maintain structural integrity and motion consistency over time. Although most recent Video DiTs have moved from discrete cross-attention to Full Attention Yang et al. (2024b) for more accurate spatial-temporal learning, we introduce Implicit Cross-Attention derived from Full Attention. ICA still retains the essence of traditional cross-attention and guides shape editing effectively. Given text embeddings $\mathbf{E} \in \mathbb{R}^{N \times d}$ and latent video tokens $\mathbf{B} \in \mathbb{R}^{M \times d}$, Full Attention mechanism first concatenates them to form a larger matrix $\mathbf{C} = [\mathbf{E}; \mathbf{B}] \in \mathbb{R}^{(N+M) \times d}$, each row of $\mathbf{C}$ can be considered as both Query ($Q$), Key ($K$), and Value ($V$). The full attention map is computed as follows:

$$\mathcal{A} = \text{Softmax}\left(\frac{\mathbf{C}\mathbf{C}^\top}{\sqrt{d}}\right) = \begin{bmatrix} \mathcal{A}_{EE} & \mathcal{A}_{EB} \\ \mathcal{A}_{BE} & \mathcal{A}_{BB} \end{bmatrix} \in \mathbb{R}^{(N+M) \times (N+M)} \tag{14}$$

We identify that the off-diagonal block $\mathcal{A}_{EB}$ or $\mathcal{A}_{BE}$ inherently encodes cross-modal interactions. Our *Implicit Cross-Attention* extracts this block of different timesteps and binarizes it into $M_t$. We mask $\Delta v_t(Z_0, c_0, c_1)$ with $M_t$ to restrain the changes in the unedited region as Eq. 15. $M_t$ can also be optionally combined with the popular SAM Kirillov et al. (2023) masks using Boolean operations.

$$\Delta v_{t,M_t}(Z_0, c_0, c_1) = M_t \odot \left[ v_{t,c_1}(\hat{Z}_t) - v_{t,c_0}(\Phi_t(Z_0)) \right] \tag{15}$$

**Target Embedding Reinforcement**. We observe that in 3D Full-Attention, the effect of text embeddings diminishes as frame length increases. This phenomenon is particularly evident in global editing tasks such as stylization. We attribute this issue to the competition between fixed-length text tokens $\mathbf{E} \in \mathbb{R}^{N \times d}$ and an increasing number of spatiotemporal tokens $\mathbf{Z} \in \mathbb{R}^{F \times H \times W \times d}$. As the video duration grows, vectors associated with stylization embeddings become increasingly sparse across frames. This sparsity may further reduce the guidance fidelity of the text embeddings. To address these challenges, we propose Embedding Reinforcement (ER) for prompt alignment:

$$\tilde{\mathbf{E}}^{(k)} = \mathbf{E} + \gamma^{(k)} \odot \mathbf{E} \tag{16}$$

where $k$ is used to locate the target embedding for editing, and its value is amplified by $\gamma + 1$. Specifically, we set $\gamma = 0.2$ for shape editing and $\gamma = 5$ for stylization. By reinforcing the embeddings, the cross-attention map is reweighted to focus on regions more relevant to the editing target, enhancing editing precision. Refer to the Appendix C for more method statements.

## 4 RESULTS

We evaluate DFVEdit on a comprehensive set of video editing tasks, including local editing (object shape and attribute modification) and global editing (style transfer) for both single (Sec. 4.1) and multiple objects editing (Appendix G). Using public videos from DAVIS2017 Pont-Tuset et al. (2017) and Pexels Pexels, our evaluation covers: (1) comparisons with state-of-the-art training-free and finetuning-based methods; (2) ablation analysis of DFVEdit's key components; and (3) extension experiments across architectures and tasks. Additional experiment results are provided in the Appendix G, and experiment setting details are provided in the Appendix D. DFVEdit outperforms zero-shot editing methods and surpasses competing fine-tuned methods Yatim et al. (2024); Wu et al. (2023); Liu et al. (2024c) in all scenarios, excelling in structural fidelity, motion integrity, and temporal consistency. Ablation studies confirm each component's contribution, demonstrating strong scalability across Video DiT architectures (Fig. 4, Fig. 3, Tab. T6) and generalizing well to 2D U-Net-based image diffusion models (Tab. T7, Fig. F7).

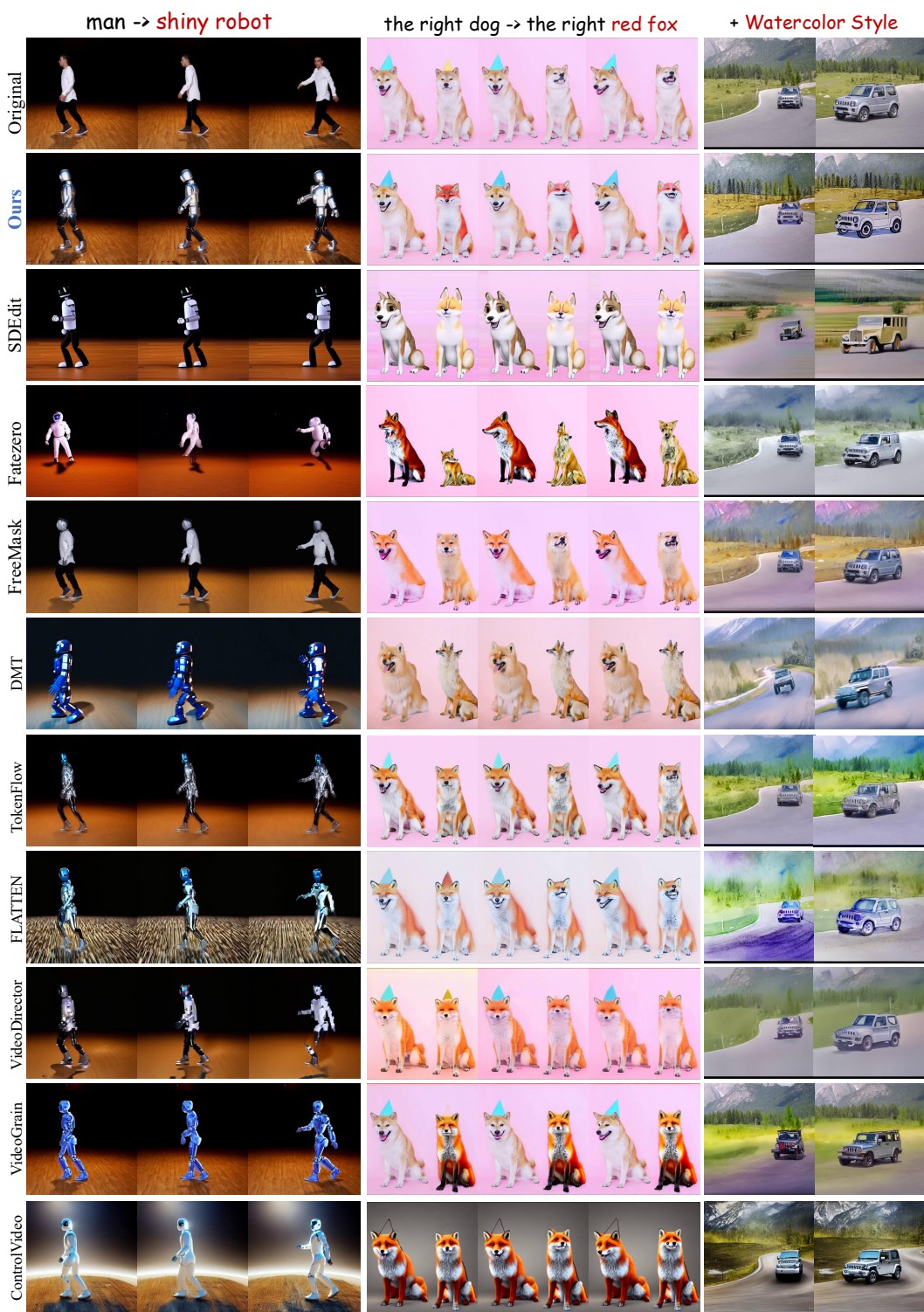

Figure 3: **Comparison.** Most methods based on attention-engineering (FateZero Qi et al. (2023),To-kenFlow Geyer et al. (2023), VideoDirector Wang et al. (2024b), FreeMask Cai et al. (2024)) suffer from flickering and fail in multi-object editing. While VideoGrain Yang et al. enhances multi-object editing, it is inferior in structure consistency and motion detail fidelity (the second column). Attention-engineering-free approaches (FLATTEN Cong et al., DMT Yatim et al. (2024), ControlVideo Zhang et al. (2023b), SDEdit Meng et al. (2021)) exhibit structural infidelity. In comparison, our method achieves SOTA performance in fidelity, alignment, and temporal consistency.

Table 1: **Quantitative evaluation and user study results.**

| Method | Consistency | | Fidelity | | Alignment | User Study | | | Computation Efficiency | | |
|---|---|---|---|---|---|---|---|---|---|---|---|
| | CLIP-F↑ | $E_{warp}$↓ | M.PSNR↑ | LPIPS↓ | CLIP-T↑ | Edit↑ | Quality↑ | Consistency↑ | VRAM↓ | RAM↓ | Latency↓ |
| SDEdit Meng et al. (2021) | 0.9811 | 1.67 | 20.52 | 0.4090 | 27.46 | 66.57 | 80.45 | 85.66 | 1.01 | 1.13 | **0.87** |
| FateZero Qi et al. (2023) | 0.9289 | 3.09 | 23.39 | 0.2634 | 26.08 | 58.87 | 50.63 | 56.89 | 2.32 | 21.44 | 3.40 |
| FreeMask Cai et al. (2024) | 0.9699 | 2.00 | 29.92 | 0.2314 | 27.06 | 75.88 | 74.67 | 77.13 | 1.64 | 25.58 | 5.65 |
| Tokenflow Geyer et al. (2023) | 0.9583 | 1.48 | 29.97 | 0.2247 | 29.78 | 70.12 | 53.45 | 57.41 | 1.43 | 3.69 | 13.03 |
| VideoDirector Wang et al. (2024b) | 0.9555 | 2.44 | 28.97 | 0.3205 | 27.50 | 74.13 | 73.25 | 71.45 | 6.00 | 2.26 | 27.97 |
| VideoGrain Yang et al. | 0.9695 | 2.68 | 30.70 | 0.2948 | 27.79 | 76.41 | 79.87 | 70.61 | 2.35 | 2.61 | 13.44 |
| FLATTEN Cong et al. | 0.9510 | 4.89 | 15.91 | 0.3559 | 27.57 | 63.45 | 69.45 | 68.32 | 1.54 | 7.31 | 4.61 |
| ControlVideo Zhang et al. (2023b) | 0.9533 | 3.10 | 10.08 | 0.4015 | 27.06 | 56.08 | 55.33 | 59.41 | 8.74 | 1.62 | 9.45 |
| DMT Yatim et al. (2024) | 0.9668 | 3.50 | 15.95 | 0.5096 | 25.34 | 62.66 | 68.36 | 69.88 | 5.64 | 3.32 | 24.40 |
| DFVEdit (on CogvideoX-5B) | **0.9924** | **1.12** | **31.18** | **0.1886** | **30.84** | **87.65** | **84.56** | **86.98** | **0.95** | **0.86** | 1.20 |
| w/o ICA | 0.9922 | 1.25 | 29.33 | 0.1920 | 31.02 | 86.45 | 84.33 | 86.56 | 0.94 | 0.78 | 1.19 |
| w/o EmbedRF | 0.9913 | 1.13 | 31.15 | 0.1889 | 29.25 | 86.04 | 83.15 | 86.13 | 0.95 | 0.85 | 1.20 |

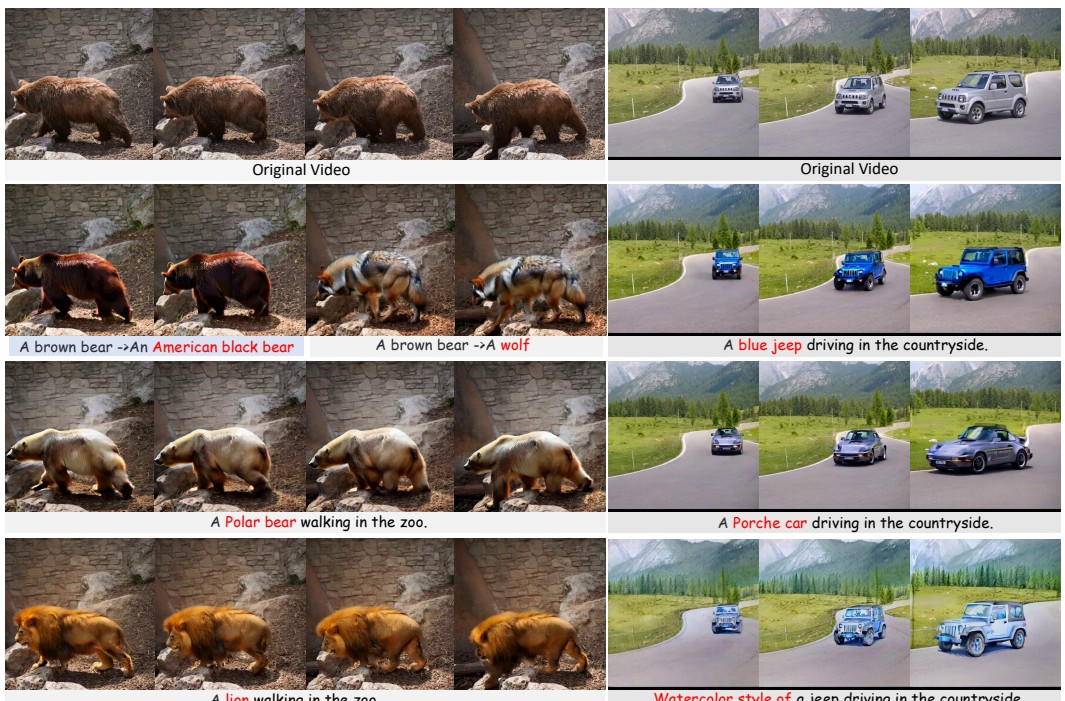

Figure 4: **Extensive qualitative results.** The extensive experiments take Wan2.1-14B Wang et al. (2025) as the base model, demonstrating the generalization of DFVEdit for Video DiTs.

## 4.1 COMPARISON RESULTS

**Qualitative evaluation.** Fig. 3 and Fig. F8 provide qualitative comparison results, showcasing our method's superiority in structure fidelity, motion integrity, and temporal consistency over other prominent baselines. For **single object editing** (first column), FateZero Qi et al. (2023), Token-Flow Geyer et al. (2023), and VideoDirector Wang et al. (2024b) exhibit noticeable flickering, while ControlVideo Zhang et al. (2023b), FLATTEN Cong et al., and finetuning-based methods DMT Yatim et al. (2024), Tune-A-Video Wu et al. (2023), and VideoP2P Liu et al. (2024c) fail to preserve the details of unedited regions. For **multi-object editing** (second column), most methods struggle with editing accuracy; although VideoGrain Yang et al. achieves success in multi-object editing using fine-grained SAM Kirillov et al. (2023) masks, it falls short in maintaining motion detail fidelity (e.g., a mismatch between the fox and dog expressions). For **stylization** (third column), Freemask Cai et al. (2024), which is based on a UNet-based video diffusion model, performs notably well, while other methods still show inconsistencies in color tone and structural details (refer to the supplementary material for video displays). Additionally, we extended FateZero Qi et al. (2023) and KVEdit Zhu et al. (2025) directly to Cogvideo-5B Yang et al. (2024b) to compare editing quality and efficiency. Due to space limitations, please refer to the appendix for more detailed comparison results. Fig. 4 provides the extensive experiment results on Wan2.1-14B Wang et al. (2025), which also demonstrates high editing quality with respect to structure fidelity, motion integrity, and prompt alignment. Wan Wang et al. (2025) is combined with Flow Matching Lipman et al. (2022), while CogVideoX Yang et al. (2024b) is based on Score Matching Song et al. (2020b). As illustrated in both Fig. 4 and Fig. 3,

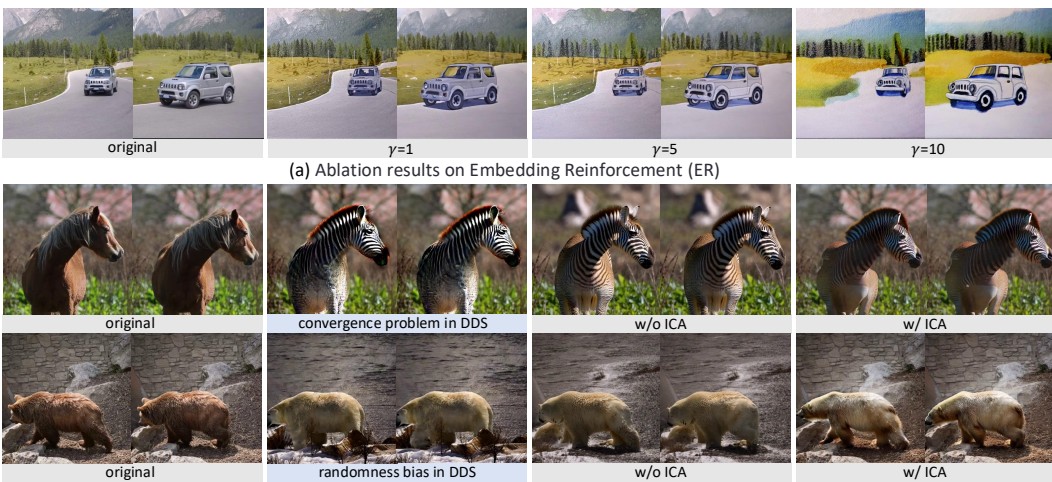

(a) Ablation results on Embedding Reinforcement (ER)

(b) Ablation results on replacing CDFV with DDS vector.

(c) Ablation results on Implicit Cross Attention (ICA) Guidance

Figure 5: **Ablation.** (a)(c) demonstrate the effectiveness of ER and ICA. (b) highlights limitations of popular approximation-based latent refinement methods Hertz et al. (2023) in video editing, including: low convergence leading to unnatural changes and unpredictable convergence times; randomness bias resulting in unsatisfactory structural fidelity.

DFVEdit achieves consistent editing quality across popular Video DiTs, whether based on Score Matching Song et al. (2020b) or Flow Matching Lipman et al. (2022).

**Quantitative Evaluation.** In Tab. 1, we compare DFVEdit with baseline methods using both automatic metrics and user study evaluations. Detailed quantitative metrics can be found in Appendix D.2, and user study details are provided in Appendix D.3. These results highlight DFVEdit's practical efficiency and efficacy. We extended FateZero Qi et al. (2023) and KVEdit Zhu et al. (2025) to CogVideoX-5B Yang et al. (2024b) to assess their performance. As shown in Fig. 1(b), these methods, originally designed for image diffusion, incur significant computational overhead when applied to Video DiTs. Refer to the Appendix E for more quantitative experiment results on insight. Tab. 1 shows DFVEdit's superior performance across multiple metrics: structural consistency CLIP-F ($0.9924$), motion fidelity $E_{warp}$ ($1.12$), unedited region fidelity M.PSNR ($31.18$), overall structural fidelity LPIPS ($0.1886$), and video-prompt alignment CLIP-T ($30.84$). User studies confirm its lead in editability ($87.65$), quality ($84.56$), and consistency ($86.98$). Additionally, it achieves notable efficiency in VRAM ($0.95$) and RAM ($0.86$) usage, with minimal latency ($1.20$).

### 4.2 ABLATION RESULTS

We evaluate the efficacy of CDFV, ICA, and ER in our ablation study. Tab. 1 reveals that omitting either the ICA or ER modules degrades performance, highlighting their indispensable roles in achieving optimal outcomes. In Fig. 5(a), we vary the Embedding Reinforcement factor $\gamma$ from 1 to 10. Without reinforcement ($\gamma = 1$), stylization effects are negligible. Stylization improves as $\gamma$ increases but degrades with excessively high values. Empirically, $\gamma = 5$ optimizes stylization without compromising structural fidelity or visual quality. Fig. 5(c) shows that omitting Implicit Cross-Attention Guidance leads to unintended changes in unedited regions. Incorporating cross-attention mechanisms significantly enhances structural fidelity and overall quality. In Fig. 5(b), we replace CDFV with the stochastic latent refinement vector in DDS Hertz et al. (2023). In this ablation, for the 'horse' experiment, ICA and ER are kept, while for the 'bear' experiment, they are omitted for a fair comparison. The results highlight the effectiveness of CDFV. For additional qualitative and quantitative comparison and ablation results, please refer to the Appendix F.

### 5 CONCLUSION

We present DFVEdit, an efficient and effective zero-shot video editing framework tailored for Video Diffusion Transformers. DFVEdit realizes video editing through the direct flow transformation of the clean source latent. We theoretically unify editing and sampling from the continuous flow perspective, propose CDFV to estimate the flow vector from the source video to the target video, and further enhance the editing quality with ICA guidance and ER mechanism. Extensive experiments demonstrate the efficacy of DFVEdit on Video DiTs.

## 6 STATEMENT

**Ethics Statement.** This work does not involve human subjects or personally identifiable information. The datasets used are publicly available and have been previously released for research purposes under appropriate licenses. Our method does not enable capabilities that are likely to cause harm when deployed, though we acknowledge that any machine learning model capable of generating realistic content could potentially be misused. No ethical approval was required for this study.

**Reproducibility Statement.** To ensure the reproducibility of our results, we include detailed experiment settings in the Appendix. Our code will be open-sourced upon publication. All datasets used in our experiments are publicly available. Implementation details for baseline methods are also provided to facilitate fair comparison.

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

# A  LLM USAGE STATEMENT

We used GPT-4 (via OpenAI's API) to assist with language polishing, grammar correction, and structural refinement of the manuscript. Specifically, the model was used to help draft and revise the introduction and related work sections, and to improve the clarity and flow of the overall writing. All prompts and generated content were carefully reviewed, critically edited, and fact-checked by the authors. The core ideas, technical content, experimental design, and final text are the responsibility of the authors and were not generated by the model.

# B  ADDITIONAL THEORETICAL DETAILS

## B.1  REVISITING VIDEO EDITING FROM SAMPLING PERSPECTIVE

Let $\{X_t^{\text{edit}}\}_{t=0}^T$ define the state trajectory of the edited video in the sampling process. We formalize video editing as a *controlled Markov chain* with the following recursive relation:

$$X_{t-1}^{\text{edit}} = g_{\theta_{2,t}}\Big(X_t^{\text{edit}},\ \underbrace{\epsilon_{\theta_1}(X_t^{\text{edit}}, t)}_{\text{Canonical Denoiser}} + \lambda \underbrace{C(X_t^{\text{edit}}, t, *)}_{\text{Control Term}}\Big) \tag{17}$$

where *State Transition* $g_{\theta_{2,t}}$ is the differentiable transition function parameterized by learnable $\theta_2$, $\epsilon_{\theta_1}$ is the pretrained diffusion model with frozen $\theta_1$, *Control Term* $C$ is the editing condition injector with intensity $\lambda \geq 0$.

The formulation maintains ***consistency with standard diffusion sampling process*** when $\lambda = 0$ and $g_{\theta_2} = \mathcal{I}$, where $\mathcal{I} : \mathcal{X} \to \mathcal{X}$ denotes the identity operator satisfying $\mathcal{I}(x) = x,\ \forall x \in \mathcal{X}$.:

$$X_{t-1}^{\text{edit}}\big|_{\substack{\lambda=0 \\ g_{\theta_2}=\mathcal{I}}} \equiv X_{t-1}^{\text{orig}} \tag{18}$$

## B.2  UNIFICATION WITH VARIOUS EDITING METHODS

Existing popular editing paradigms emerge as special cases of our control framework:

1. **Inversion-based editing** (like Fatezero Qi et al. (2023)):

$$g_{\theta_{2,t}}(a, b) = \frac{\sqrt{\alpha_{t-1}}}{\sqrt{\alpha_t}}(a + \Delta\beta_t b) \tag{19}$$

$$C(X_t^{\text{edit}}, t, *) = \epsilon_{\theta_1}^{\text{edit}}(X_t^{\text{edit}}, t) - \epsilon_{\theta_1}(X_t^{\text{edit}}, t) \tag{20}$$

$$\Delta\beta_t = \sqrt{\frac{1 - \alpha_{t-1}}{\alpha_{t-1}}} - \sqrt{\frac{1 - \alpha_t}{\alpha_t}} \tag{21}$$

2. **Latent-approximation-based editing** (like DDS Hertz et al. (2023)):

$$g_{\theta_{2,t}}(a, b) = \text{Proj}_{\theta_{2,t}}(a + \eta b) \tag{22}$$

$$C(x_t, t, *) = \epsilon_{\theta_1}(x_t, t) - \epsilon_{\theta_1}(x_t, t) - \epsilon \tag{23}$$

$$\epsilon \sim \mathcal{N}(0, \sigma_t^2 I) \tag{24}$$

where $\alpha_t$ is the DDPM noise schedule coefficient at step $t$, $\Delta\beta_t$ is the noise scale difference term maintaining consistency in the reverse process, $\epsilon_{\theta_1}^{\text{edit}}$ is the edited noise prediction conditioned on the target prompt, $X_t^{\text{edit}}$ is the latent representation during the editing process. And $Proj_{\theta_{2,t}}$ is a shallow approximation network with learnable parameter $\theta_2$ that directly refines the latent to the target latent, $\eta$ is the step size controlling parameter update strength, $\sigma_t$ is the time-dependent noise scale for stochastic refinement, and $\epsilon$ is the Gaussian noise enabling exploration in the latent space. These formulations show how various editing methods are implicitly isomorphic with the sampling process.

### B.3 REVISITING VIDEO EDITING FROM THE CONTINUOUS FLOW TRANSFORMATION PERSPECTIVE

For the sampling process of SDE:

$$dX = \underbrace{-\frac{1}{2}\beta(t)X}_{f(X,t)} dt + \underbrace{\sqrt{\beta(t)}}_{g(t)} dW \tag{25}$$

For the inverse process of SDE:

$$dX = \left[-\frac{1}{2}\beta(t)X - \beta(t)\nabla_X \log p_t(X)\right] dt + \sqrt{\beta(t)}dW \tag{26}$$

when changing the discrete update formulation into continuous $\Delta t \to 0$, we define:

$$\alpha(t) = e^{-\int_0^t \beta(s)ds}, \quad \sigma(t) = \sqrt{\frac{1-\alpha(t)}{\alpha(t)}} \tag{27}$$

Using Taylor's expansion, we have:

$$\alpha_{t-1} \approx \alpha(t) - \dot{\alpha}(t)\Delta t \tag{28}$$

$$\frac{\sqrt{\alpha_{t-1}}}{\sqrt{\alpha_t}} \approx 1 - \frac{1}{2}\beta(t)\Delta t \tag{29}$$

$$X_{t-1}^{\text{edit}} \approx X_t^{\text{edit}} - \frac{\beta(t)}{2}\left(X_t^{\text{edit}} + \sigma(t)(\epsilon_{\theta_3} + \lambda C)\right)\Delta t \tag{30}$$

Under the Euler discretization scheme with step size $\Delta t \to 0$ and $g_{\theta_2} = \mathcal{I}$, the discrete process equation 17 converges to the controlled SDE:

$$dX_t^{\text{edit}} = \underbrace{\left[-\frac{\beta(t)}{2}X_t^{\text{edit}} + \frac{\beta(t)}{2}\nabla \log p_t(X_t^{\text{edit}}) + \lambda\frac{\beta(t)}{2}\sigma(t)C(X_t^{\text{edit}},t)\right]}_{f_{\theta_1}(X_t^{\text{edit}},t)} dt + \underbrace{\sqrt{\beta(t)}}_{g(t)} dW \tag{31}$$

And our derived CDFV adheres to the *minimum intervention principle* from optimal control theory, which theoretically guarantees computational efficiency:

$$\min_{\lambda,C} \mathbb{E}\left[\int_0^T \|C(X_t,t)\|^2 dt\right] \quad \text{s.t.} \quad dX = \left[f_{\theta_1}(X,t) + \lambda C(X,t)\right] dt + g(t)dW \tag{32}$$

$$C^*(X,t) = \frac{\nabla_X \log p_t^{\text{edit}}(X) - \nabla_X \log p_t(X)}{\sigma(t)} \tag{33}$$

In addition, we provide the simplified algorithm of DFVEdit as below:

## C  INNOVATIONS OVER TRADITIONAL ATTENTION-ENGINEERING APPROACHES

### C.1  CONCEPTUAL AND OPERATIONAL DISTINCTIONS

- **Difference on the conceptual foundation**: We unify editing and sampling from the flow perspective, and model editing as a *continuous flow transformation* in latent space via CDFV, whereas attention engineering manipulates feature correlations via Q/K/V modifications (caching, replacing, or fusing).

---

**Algorithm 1** Simplified algorithm for DFVEdit

---

**Require:** source video $\mathbf{X}_0$, target and source prompt embeddings $[C_1, C_0]$, Video DiT $\epsilon_{\theta_1}$, encoder $\mathcal{E}(\cdot)$, decoder $\mathcal{D}(\cdot)$, sampling timesteps $T$, ER scale $\gamma^{(k)}$

**Ensure:** Edited video $\mathbf{X}_1$

1: $\mathbf{Z}_0 \leftarrow \mathcal{E}(\mathbf{X}_0)$ ▷ Latent encoding
2: $\hat{\mathbf{Z}}_T \leftarrow \mathbf{Z}_0$ ▷ Initialize target latent
3: $\tilde{C}_1 \leftarrow C_1 + \gamma^{(k)} \odot C_1$ ▷ Embedding Reinforcement
4: **for** $t \leftarrow T$ **down to** $1$ **do**
5: $\quad \mathbf{Z}_{\text{trans}} \leftarrow \Phi_t([\hat{\mathbf{Z}}_t; \mathbf{Z}_0])$ ▷ One-step forward process: $q(\mathbf{z}_t|\mathbf{z}_0)$
6: $\quad \Delta v_t \xleftarrow{v_{(t,c_1)}, v_{(t,c_0)}}{\Delta} \epsilon_{\theta_1}(\mathbf{Z}_{\text{trans}}, [\tilde{C}_1, C_0])$ ▷ Raw CDFV prediction
7: $\quad \Delta v_{(t,M_t)} \leftarrow M_t \odot [\Delta v_t]$ ▷ Implicit Cross-Attention Guidance
8: $\quad \hat{\mathbf{Z}}_{t-1} \leftarrow \hat{\mathbf{Z}}_t - \Delta v_{(t,M_t)}$ ▷ Latent update
9: **end for**
10: $\mathbf{X}_1 \leftarrow \mathcal{D}(\hat{\mathbf{Z}}_0)$ ▷ Video synthesis

---

**Note**:
▷ Flow map $\Phi_t$ implements the one-step forward process with standard method-specific coefficients (DDPM/DDIM/Flow Matching).
▷ ICA mask $M_t$ is computed from the specific layer of the Full Attention map (Section 3.3).

---

- **Difference on the operational mechanism**: CDFV performs end-to-end deformation, and ICA only infers an implicit mask from cross-attention or full-attention maps to constrain CDFV's spatial influence, and SAM masks are also applied to CDFV optionally (used in multi-object shape editing), which means ICA and SAM masks are operated on the latent space without engaging in traditional attention engineering.

- **Difference on the architectural impact**: DFVEdit's flow-based formulation enables *model-agnostic operation* (has been verified on 2D U-Net and various Video DiTs), while most attention engineering works fail on Video DiTs due to architectural and computational constraints.

## C.2 ADVANTAGES OF ICA OVER TRADITIONAL ATTENTION ENGINEERING

ICA does not modify attention weights or feature activations but instead infers an implicit mask from attention responses to guide the CDFV. This fundamental distinction brings several key advantages:

- **Preservation of Feature Distribution**: By not altering attention outputs, ICA avoids unintended distribution shifts in unedited regions.

- **Computational Efficiency**: The mask is computed once per timestep via a forward pass, contrasting with the iterative recomputation required by attention engineering.

- **Architecture Compatibility**: As a lightweight component of CDFV, ICA can seamlessly deploy across various architectures, including 2D U-Net and Video DiTs, where traditional attention editing faces integration challenges.

## C.3 USE OF SAM MASKS IN DFVEDIT

DFVEdit uses SAM masks selectively for enhancing multi-object editing scenarios, particularly when dealing with semantic leakage issues like distinguishing between left/right objects in Fig. F10. Unlike other methods that require precise SAM masks as essential input for operation, DFVEdit operates primarily without SAM. Spatial constraints are mainly derived from ICA, producing *timestep-aware, coarse-to-fine* masks. When used, SAM masks undergo padding to soften edges and intersect with ICA maps, ensuring diffusion-aligned, soft masks that enhance editing fluidity and naturalness.

## D EXPERIMENTAL SETTINGS

### D.1 DATASETS

We evaluate our method on the public DAVIS2017 dataset Pont-Tuset et al. (2017) and open-source videos from Pexels Pexels, following the video editing benchmarks established by popular methods such as FateZero Qi et al. (2023) and TokenFlow Geyer et al. (2023). All experiments are conducted on one A100-80G GPU.

All experiments are conducted on 40-frame video sequences at a resolution of $512 \times 512$. Our focus is on training-free appearance editing, encompassing both *local editing* (e.g., shape and attribute modification) and *global editing* (e.g., stylization). To comprehensively assess motion preservation capabilities, we design experiments spanning four categories of motion complexity and articulation level: *1) High Dynamics, Low Articulation:* e.g., jeep stylization (Fig. 3) – characterized by large-scale motion with minimal structural deformation. *2) Moderate Dynamics, Moderate Articulation:* e.g., human-to-robot translation (Fig. 3, Fig. F12) – involving intermediate limb articulation and pose transitions. *3) Low Dynamics, High Articulation:* e.g., dogs-to-foxes translation (Fig. 3, Fig. F12, Fig. F10) – requiring preservation of fine, high-frequency non-rigid motions (e.g., ear flicks, blinking, facial expressions). *4) High Dynamics, High Articulation:* e.g., cyclist editing (Fig. F10, Fig. F11) – combining fast motion, complex limb articulation, and micro-action transfer.

Notably, while competing methods are typically limited to editing short video clips (under 20 frames) due to memory constraints (Tab. T3), our approach demonstrates robust performance on extended 40-frame sequences without memory overflow or significant distortion. Additionally, as evidenced in Fig. F10 and Fig. 3, our method exhibits superior capability in handling complex multi-object editing scenarios. Given the inherent limitations of static figures for conveying temporal dynamics, we strongly encourage readers to consult our supplementary video. This resource provides clear demonstrations of temporal coherence, editing fidelity, and motion preservation across all evaluated scenarios.

### D.2 QUANTITATIVE METRICS

We do quantitative and human evaluations with 8 quantitative metrics, including Temporal Consistency ('CLIP-F'), Warping Error ('$E_{warp}$') Geyer et al. (2023), Prompt Alignment ('CLIP-T'), Masked PSNR ('M.PSNR'), Perceptual Similarity ('LPIPS'), Relative GPU Memory Consumption ('VRAM'), Relative CPU Memory Consumption ('RAM'), Relative Inference Latency ('Latency') and 3 metrics for user study, including Text Alignment ('Edit'), Overall Frame Quality ('Quality') and Temporal Consistency and Realism ('Consistency'). Specifically, **CLIP-F** calculates inter-frame cosine similarity to assess structural consistency, while **$E_{warp}$** measures warping error Geyer et al. (2023) to evaluate motion fidelity. Additionally, **M.PSNR** computes the Masked Peak Signal-to-Noise Ratio between source and target videos to gauge the fidelity of unedited regions, and **LPIPS** evaluates the Learned Perceptual Image Patch Similarity for overall structural fidelity. Moreover, **CLIP-T** quantifies the alignment between the target prompt and video through the CLIP Score Hessel et al. (2021). The results demonstrate that DFVEdit achieves superior spatial-temporal consistency, fidelity, and prompt alignment compared to other methods. Furthermore, to evaluate memory and computational efficiency, we measure Relative GPU Memory Consumption (**VRAM**), defined as the ratio of editing consumption on GPU relative to original inference consumption; Relative Inference Latency (**Latency**), which assesses the ratio of editing latency to inference latency; and Relative CPU Memory Consumption (**RAM**), measuring the ratio of editing consumption on CPU over original inference consumption.

**CLIP metrics.** We employ the output logits of the official ViT-L-14 CLIP model to compute two metrics: (1) the mean cosine similarity between all frame embeddings and the target text prompt (CLIP-T), and (2) the average cosine similarity of consecutive frame embeddings of edited videos (CLIP-F).

**Masked PSNR.** We quantify structural preservation by computing Masked PSNR on unedited regions, following Liu et al. (2024b). Using 10 DAVIS Pont-Tuset et al. (2017) videos, 30 diverse prompts and corresponding segmentation annotations $M$ provided by DAVIS, we calculate pixel-level differences between source ($\mathbf{X}_0$) and edited ($\mathbf{X}_1$) videos within regions identified by inverted segmentation masks $M^* = \neg M$.

$$M.PSNR(\mathbf{X_1}, \mathbf{X_0}) = PSNR(B(\mathbf{X_1}, M^*), B(\mathbf{X_1}, M^*)) \tag{34}$$

where $B(\dots)$ is the binary operation with a threshold of 0.3.

**Relative Efficiency Metrics.** We evaluate computational efficiency through three normalized metrics: Relative CPU Memory Consumption means the ratio of average CPU Memory allocated during editing to that average CPU allocated to original inference (only generation) with the same base model. Relative Inference Latency means the ratio of the latency with editing to that of the original inference latency with the same model. These relative metrics enable fair comparison across varying base model requirements. Tab. T2 reports absolute values and experimental configurations. Reported values include absolute peak allocated GPU/CPU memory consumption (MB), processing latency, frame count ($F$), and corresponding base models. The groups 'Stable Diffusion', 'Zeroscope', and 'CogVideoX' represent the original generation results with base models, while other groups are the editing results.

Table T2: **Absolute empirical computational efficiency results.**

| Method | GPU Memory (MB) | CPU Memory (MB) | Latency (s) | $F$ | Base Model |
|---|---|---|---|---|---|
| Stable Diffusion Rombach et al. (2022) | 4134.31 | 2865.00 | 15.47 | 8 | Stable Diffusion 1.5 |
| Zeroscope Wang et al. (2023) | 4551.32 | 3833.86 | 13.14 | 8 | Zeroscope |
| CogVideoX Yang et al. (2024b) | 33110.36 | 10522.26 | 100.80 | 41 | CogVideoX-5B |
| SDEdit Meng et al. (2021) | 33441.46 | 11890.15 | 87.69 | 41 | CogVideoX-5B |
| FateZero Qi et al. (2023) | 9576.13 | 61416.08 | 52.58 | 8 | Stable Diffusion 1.5 |
| FreeMask Cai et al. (2024) | 7464.16 | 98059.40 | 74.29 | 8 | Zeroscope |
| TokenFlow Geyer et al. (2023) | 17040.29 | 159475.48 | 126.87 | 8 | Stable Diffusion 1.5 |
| VideoDirector Wang et al. (2024b) | 24789.38 | 6475.66 | 432.64 | 8 | Stable Diffusion 1.5 |
| FLATTEN Cong et al. | 6385.01 | 20936.98 | 71.35 | 8 | Stable Diffusion 1.5 |
| ControlVideo Zhang et al. (2023b) | 36115.23 | 4635.36 | 146.19 | 8 | Stable Diffusion 1.5 |
| DMT Yatim et al. (2024) | 42500.24 | 25572.34 | 217.54 | 8 | Stable Diffusion 1.5 |
| Ours | 31454.84 | 9049.14 | 120.96 | 41 | CogVideoX-5B |

### D.3 USER STUDY

Regarding user studies, we focus on Target Prompt Alignment (**Edit**), Overall Editing Quality including fidelity of unedited areas, minimal filtering and blurring (**Quality**), and Motion and Structural Consistency (**Consistency**). We conducted a pairwise comparison study with 20 participants evaluating 80 video-prompt pairs (30 from DAVIS, 50 from the website Pexels Pexels). Participants rated three aspects: (1) *Text Alignment* (prompt-video correspondence), (2) *Frame Quality* (visual artifacts), and (3) *Consistency* (temporal coherence and motion preservation). Scores (0-100 scale) were aggregated by trimming extremes and averaging remaining responses, yielding 1600 total ratings.

### D.4 BASE MODELS

DFVEdit is focused on fully exploiting the capabilities of Video DiTs for high-quality zero-shot video editing. We take CogVideoX-5B Yang et al. (2024b) and Wanx2.1-14B Wang et al. (2025) as our base models. To the best of our knowledge, DFVEdit is the first method enabling efficient and effective zero-shot video editing on modern Video DiTs—including both score-matching (e.g., CogVideoX) and flow-matching (e.g., Wan2.1) models. Prior attention-based methods are often incompatible due to architectural and computational constraints. DFVEdit overcomes these via a lightweight, flow-field-based formulation. For a fair comparison and to show DFVEdit's model-agnostic operation ability, we also conduct experiments on a unified 2D U-Net (Stable Diffusion 1.5) for image editing (Fig. F7, Tab. T7), demonstrating the effectiveness of DFVEdit independent of Video-DiT-specific advantages.

### D.5 BASELINES

For baselines, we compare against training-free editing methods, including FateZero Qi et al. (2023), TokenFlow Geyer et al. (2023), VideoDirector Wang et al. (2024b), FreeMask Cai et al. (2024), VidToMe Li et al. (2024a) and VideoGain Yang et al., which rely on attention engineering; ControlVideo Zhang et al. (2023b), FLATTEN Cong et al., which are free of attention engineering; and

SDEdit Meng et al. (2021) (directly applied to CogVideoX-5B Yang et al. (2024b) base model for video editing), CogVideoX-V2V Yang et al. (2024b), as well as finetuning-based method DMT Yatim et al. (2024), Tune-A-Video Wu et al. (2023), Video-P2P Liu et al. (2024c).

## E   EXPERIMENTAL DETAILS ON INSIGHT

In Fig. 1 of the main text, we present some visualizations of our insights: (a) presents the theoretical attention memory consumption of different base models; (b) presents the theoretical inference latency of FateZero Qi et al. (2023) and KVEdit Zhu et al. (2025) when directly applying them to CogVideoX-5B, as well as the practical inference latency of DFVEdit on CogVideoX-5B. (c) reveals the evolution pattern of DFV from coarse contours to fine details, consistent with the diffusion sampling process, providing an intuitive motivation for a unified perspective on editing and generation. Here, we provide more details on these insights.

**Attention memory explosion in DiT models.** We analyze the memory consumption of attention mechanisms in the Unet module of diffusion models versus the Transformer module of DiT models, focusing on estimating the memory (GB) needed for storing attention score maps in float32 format per timestep. Although attention score maps are rarely computed explicitly in base models due to efficiency concerns, traditional editing methods often require their direct manipulation for attention engineering. Therefore, explicit examination of these maps helps in identifying challenges in adapting traditional editing techniques to modern DiT architectures. While our analysis centers on the memory footprint of attention score maps within a single timestep, editing methods based on attention engineering may involve caching or modifying attention maps across multiple timesteps, and we highlight the significant computation overhead when applying attention-engineering-based video editing methods to Video DiTs. As shown in Tab. T3, traditional diffusion models (e.g., Stable Diffusion and Zeroscope) exhibit multi-scale attention mechanisms with shapes varying by layer. In contrast, modern Video DiTs like CogVideoX-5B employ fixed large-scale attention ([2, 48, 11490, 11490]), resulting in 283× higher memory than SD's maximum (7 GB vs. 1871 GB). This fundamental architectural shift explains the inefficiency of attention-based editing methods when applied to Video DiTs.

Table T3: **Peak attention memory consumption (GB) for full score maps (float32) per timestep.** Values with $\sim$ approximate ground truth with $\pm 5\%$ variance. $F$: processed frames. Asterisk (*) indicates dynamic attention shapes.

| Model | Attention Shape | Block Number | Attention Memory (GB) | Dtype | $F$ |
|---|---|---|---|---|---|
| Stable Diffusion Rombach et al. (2022)* | Multi-scale | 32 | ~7 | float32 | 1 |
| HunyuanDiT Li et al. (2024b) | [2,4096,4096] | 80 | ~10 | float32 | 1 |
| Zeroscope Wang et al. (2023)* | Multi-scale | 64 | ~25 | float32 | 8 |
| HunyuanVideo Kong et al. (2024) | [1,24,11520,11520] | 48 | ~612 | float32 | 41 |
| Wanx2.1-14B Wang et al. (2025) | [1,40,11264,11264] | 40 | ~794 | float32 | 41 |
| CogVideoX-5B Yang et al. (2024b) | [2,48,11490,11490] | 40 | ~1871 | float32 | 41 |

**Inference Latency Comparison.** We adopt a theoretical estimation approach to evaluate the inference latency for attention-engineering-based editing methods (FateZero Qi et al. (2023) and KVEdit Zhu et al. (2025)) and measure the practical inference latency of DFVEdit, since direct empirical testing of FateZero and KVEdit with Video DiT is infeasible due to GPU memory and CPU RAM constraints. First, we conduct performance analysis assuming unlimited CPU memory. Specifically, we measure execution times per timestep and extrapolate to the total timesteps required for editing. Given that caching attention maps, keys, and queries exceeds GPU capacity, all caching operations utilize CPU memory. This methodology provides theoretical latency estimates for these methods in Video DiT contexts. The selected approaches demonstrate that both traditional diffusion-based methods and image DiT-based methods face significant resource overheads when directly applied to Video DiTs.

## F   EXPERIMENT RESULTS ON HYPERPARAMETERS

We used the full version of DFVEdit, including both the ICA and ER components. The hyperparameters of DFVEdit are stable and easy to tune. In practice, DFVEdit introduces only a few key

hyperparameters: the fusion timesteps for ICA, the number of sampling iterations (T), and ER's hyperparameter $\gamma$.

**Seed Sensitivity and Reproducibility.** The latent initialization stage is noise-free, meaning we transform a clean source video latent into a clean target latent. When computing the Conditional Delta Flow Vector (CDFV), the same Wiener process $\epsilon$ (i.e., the same random seeds) is applied to both $z_t^t$ (target latent at $t$) and $z_t^s$ (source latent) in the forward flow (Eq. 1), which leads to the cancellation of stochastic noise in the output difference (Eq. 12). Additionally, each forward pass uses a new random seed, preventing bias accumulation from fixed noise patterns. We tested DFVEdit (on CogVideoX-5B) on 20 video-prompt pairs with five different repetitive experiments to evaluate stability across random seeds. Tab. T4 reports the mean and standard deviation of key metrics, indicating low variance and consistent performance. The stability arises from our CDFV design—only the deterministic score difference drives editing, and ICA further enhances consistency by anchoring background features.

Table T4: Performance stability across 5 repetitive experiments (mean $\pm$ std).

| Method | CLIP-F $\uparrow$ | $E_{\text{warp}} \downarrow$ | M.PSNR $\uparrow$ | LPIPS $\downarrow$ | CLIP-T $\uparrow$ |
|---|---|---|---|---|---|
| DFVEdit | $0.9924 \pm 0.003$ | $1.11 \pm 0.12$ | $31.2 \pm 1.3$ | $0.189 \pm 0.012$ | $30.8 \pm 0.9$ |

**Ablation Study on Number of Diffusion Steps** We thank the reviewer for this suggestion. To analyze the sensitivity of DFVEdit the total number of diffusion steps ($T$), we conducted an ablation study on Wan2.1-14B. As shown in Tab. T5, both editing accuracy (measured by CLIP-T $\uparrow$) and temporal consistency ($E_{\text{warp}} \downarrow$, CLIP-F $\uparrow$) stabilize when $T \geq 50$. For $T < 50$, insufficient flow field estimation leads to under-editing, manifesting as blurred or temporally unstable results. In contrast, increasing $T$ beyond 50 yields diminishing improvements at the cost of higher computational overhead, and may result in a slight decrease in performance on background preservation. We therefore set $T = 50$ as the default, achieving an optimal trade-off between efficiency and quality. This value is also consistent with common practice in high-fidelity video generation, where moderate step counts suffice under advanced solvers.

Table T5: Ablation study on the number of diffusion steps $T$.

| Total Timesteps $T$ | CLIP-F $\uparrow$ | $E_{\text{warp}} \downarrow$ | M.PSNR $\uparrow$ | LPIPS $\downarrow$ | CLIP-T $\uparrow$ |
|---|---|---|---|---|---|
| 10 | 0.9731 | 2.1305 | 30.56 | 0.1793 | 29.91 |
| 20 | 0.9833 | 2.1100 | 30.61 | 0.1688 | 30.66 |
| 50 (default) | 0.9950 | **1.0600** | **31.23** | **0.1568** | 31.34 |
| 100 | 0.9954 | 1.4175 | 30.96 | 0.1569 | 31.32 |
| 150 | 0.9973 | 1.3881 | 30.73 | 0.1571 | 31.36 |
| 200 | **0.9953** | 1.3672 | 30.66 | 0.1578 | **31.50** |

**Embedding Reinforcement.** Due to space limitations, we only visualize the embedding reinforcement ablation results on stylization in the main text; here, we additionally visualize the results on shape editing. As shown in Fig. F9, at $\gamma = 0$, the distinctive traits of polar bears compared to brown bears, such as their white fur and rounded ears, are effectively captured with high background fidelity. Increasing $\gamma$ to 1 enhances editing quality, more accurately portraying the polar bear's elongated neck and smaller head-to-body ratio. However, at $\gamma = 5$, there is a notable decline in video synthesis quality, characterized by visible flickering and noise, alongside reduced background preservation. Our experiments demonstrate that for optimal editing outcomes in shape modification, the ER method's hyperparameter $\gamma$ should be set within the range of 0 to 1. For simplicity and efficiency, we typically set $\gamma$ to 0.3 in our studies, although values within this range generally yield satisfactory results.

**On ICA masks.** Our method, though not specifically designed for multi-object editing, inherently adapts to such tasks through the editing region localization capability of CDFV. We enhance the editing precision by combining: (1) Implicit Cross-Attention (ICA) derived from Layer 16 transformer blocks (based on the observation that cross-attention masks exhibit a coarse-to-fine change across denoising timesteps as found in FreeMask Cai et al. (2024), with the layer index selection method also following FreeMask), and (2) SAM masks with edge padding. Fig. F6 shows an example of ICA extraction and visualization. This strategy operates in two phases: ICA guidance during early

Table T6: Quantitative comparison with additional state-of-the-art video editing methods.

| Method | CLIP-F↑ | E_warp↓ | M.PSNR↑ | LPIPS↓ | CLIP-T↑ |
|--------|---------|---------|---------|--------|---------|
| VideoP2P Liu et al. (2024c) | 0.9624 | 3.28 | 17.38 | 0.4531 | 27.26 |
| Tune-A-Video Wu et al. (2023) | 0.9612 | 3.35 | 16.67 | 0.4545 | 27.14 |
| CogvideoX-V2V Yang et al. (2024b) | 0.9812 | 1.67 | 20.53 | 0.4092 | 27.46 |
| VidToMe Li et al. (2024a) | 0.9737 | 2.96 | 22.52 | 0.3062 | 27.32 |
| **DFVEdit (CogvideoX-5B)** | **0.9924** | **1.12** | **31.18** | **0.1886** | **30.84** |
| **DFVEdit (on Wan2.1-14B)** | **0.9950** | **1.06** | **31.23** | **0.1568** | **31.34** |

denoising ($t = T \rightarrow 0.4T$) preserves shape flexibility while reducing background leakage, followed by SAM-based mask guidance ($t = 0.3T \rightarrow 0$).

## G ADDITIONAL EXPERIMENTAL RESULTS

**Extended Comparison Experiments.** To ensure a comprehensive evaluation, we additionally compare with CogVideoX-V2V Yang et al. (2024b) on the Video DiT backbone (CogVideoX-5B), state-of-the-art fine-tuning-based editing methods including Tune-A-Video Wu et al. (2023), VideoP2P Liu et al. (2024c), and the latest competing method VidToMe Li et al. (2024a). As illustrated in Tab. T6 and Fig. F8, DFVEdit demonstrates superior performance. Furthermore, qualitative results using Wan2.1-14B are provided in Figs. 4 and F12, with corresponding quantitative metrics in Tab. T6. These results confirm that DFVEdit achieves enhanced outcomes when leveraging this advanced Video DiT backbone.

**Image Editing Experiments** We conducted experiments on PIE-Bench Ju et al. (2023) using Stable Diffusion 1.5, comparing DFVEdit with Instruct-pix2pix Brooks et al. (2023), PnP Tumanyan et al. (2023), and P2P Hertz et al. (2022) (PnP and P2P have adopted DirectInversion Ju et al. (2023) techniques). We evaluated DFVEdit and T2I baselines on two representative subsets: *1_change_object_80* and *9_change_style_80*, using standard metrics—Distance (overall structure coherence), PSNR/LPIPS/SSIM (background preservation), and CLIP Similarity (whole alignment)—following DirectInversion. Results in Tab. T7 and Fig. F7 show the competitive performance and practical cross-modal generalization ability of DFVEdit. Our core contribution remains enabling efficient, zero-shot editing on modern Video DiT models, where DFVEdit overcomes the computational and architectural limitations of attention-based methods without requiring fine-tuning.

Table T7: Quantitative results on PIE-Bench Ju et al. (2023) for image editing.

| Method | Structure | | Background | | |
|--------|-----------|--|------------|--|--|
| | Distance ↓ | PSNR ↑ | LPIPS ↓ | SSIM ↑ | CLIP Similarity ↑ |
| DFVEdit | **0.0167** | **23.24** | **0.1225** | **0.6935** | **29.62** |
| Instruct-pix2pix Brooks et al. (2023) | 0.0258 | 21.56 | 0.1308 | 0.6321 | 26.13 |
| P2P Hertz et al. (2022) | 0.0186 | 22.38 | 0.1257 | 0.6416 | 29.07 |
| PnP Tumanyan et al. (2023) | 0.0279 | 18.06 | 0.1353 | 0.6135 | 29.02 |

**On multi-objects editing.** As shown in Fig. F10, our method demonstrates robust multi-object editing capabilities across diverse scenarios, achieving target object accuracy while maintaining non-edited region fidelity. The framework handles both complex dynamic interactions with multiple objects in cluttered environments, and fine-grained editing requiring precise motion retention. These findings not only underscore the robustness and effectiveness of our proposed method but also lay a solid foundation for future advancements in multi-object editing.

**More results on attribute editing.** Due to space limitations, we include the visualization results of attribute editing in the Appendix. As shown in Fig. F11, our method demonstrates satisfactory performance on attribute editing, enabling the natural and seamless integration of both added and removed small objects within existing scenes.

**More results on extension experiments.** We have demonstrated additional results of applying DFVEdit to the Wan2.1 base model in the main text. To further objectively evaluate the generality and robustness of DFVEdit on Video DiTs, Fig. F12 compares the performance of the same video

editing tasks across different base models using our method. This comparison aims to reveal the variations in outcomes due to differences in models and to verify the robustness and generality of our approach. The experimental results indicate that although different base models may lead to slight variations in the editing outcomes, overall, all edited videos align well with the target prompt, and the editing quality meets the expected standards. These findings reflect the high robustness and generalization ability of our proposed method. ***Refer to the 'DFVEdit.mp4' in the supplementary material for the dynamic video display. The code will be public upon publication of this work.***

## H    LIMITATIONS AND FUTURE WORK

As illustrated in Fig. F13, our method exhibits several limitations common to zero-shot video editing. First, maintaining perfect detail fidelity in non-edited regions remains challenging. While our approach demonstrates superior fidelity preservation compared to existing methods, some detail loss persists even with ICA guidance. This issue is partially attributable to VAE compression artifacts, a fundamental bottleneck in latent diffusion models that particularly affects high-frequency textures and fine structures. In addition, our method has limited capability for large shape variations (e.g., bicycle-to-car transformations). The flow-based formulation and the underlying DiT's structural dependencies restrict applications to shape editing tasks with minor layout changes. Generating entirely new geometries while preserving background consistency and motion dynamics remains an open challenge in video editing. Furthermore, DFVEdit is primarily designed for appearance editing and currently offers limited support for non-rigid shape modifications Yoon et al. (2024a) or drag-based interactive editing Teng et al. (2023); Deng et al. (2024). Future work could explore incorporating explicit structural guidance (e.g., depth, optical flow) or layered compositing strategies to address these limitations, potentially enabling more extreme shape transformations and interactive editing capabilities.

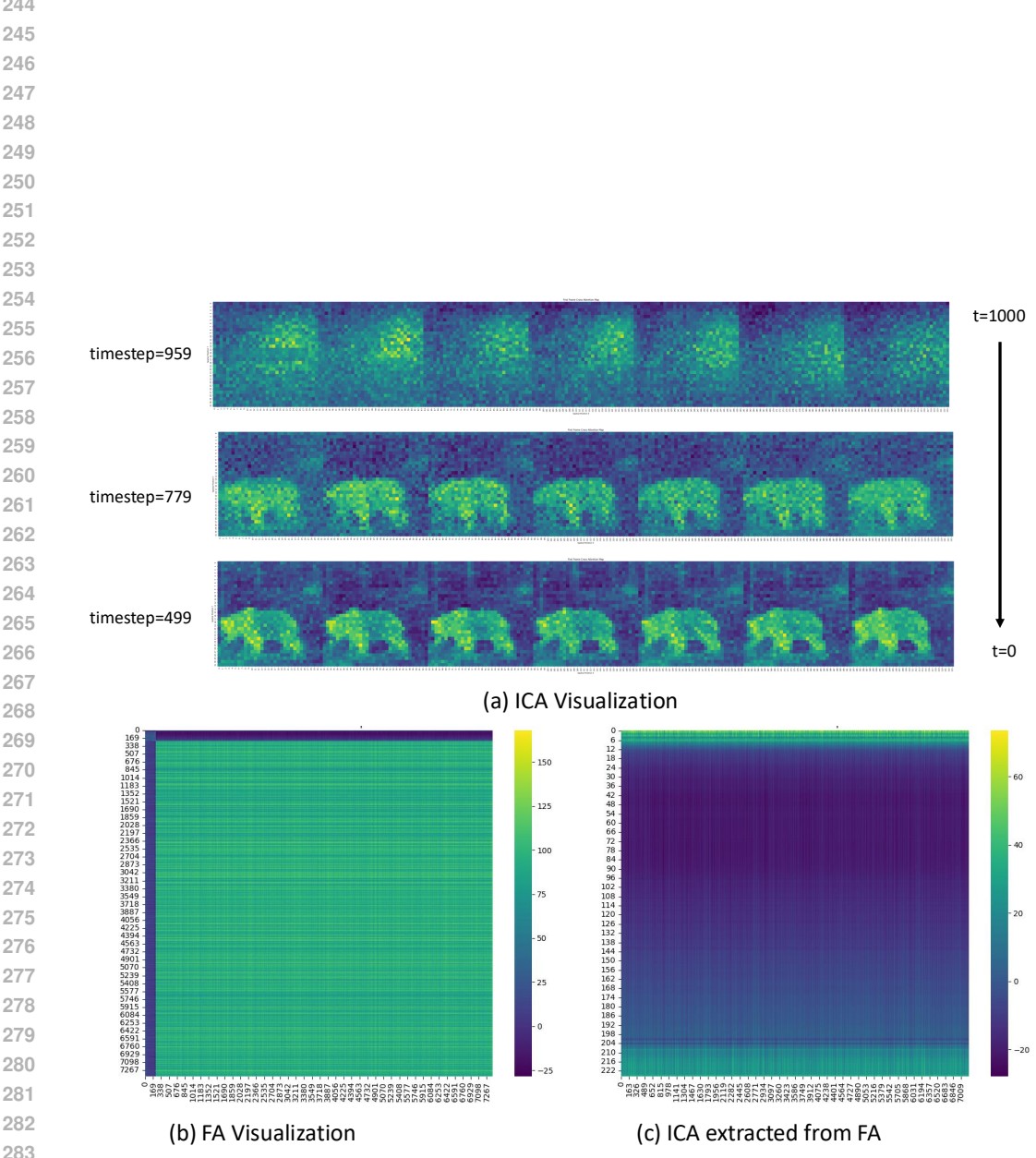

(a) ICA Visualization

(b) FA Visualization

(c) ICA extracted from FA

Figure F6: **Implicit Cross Attention extraction and visualization.**

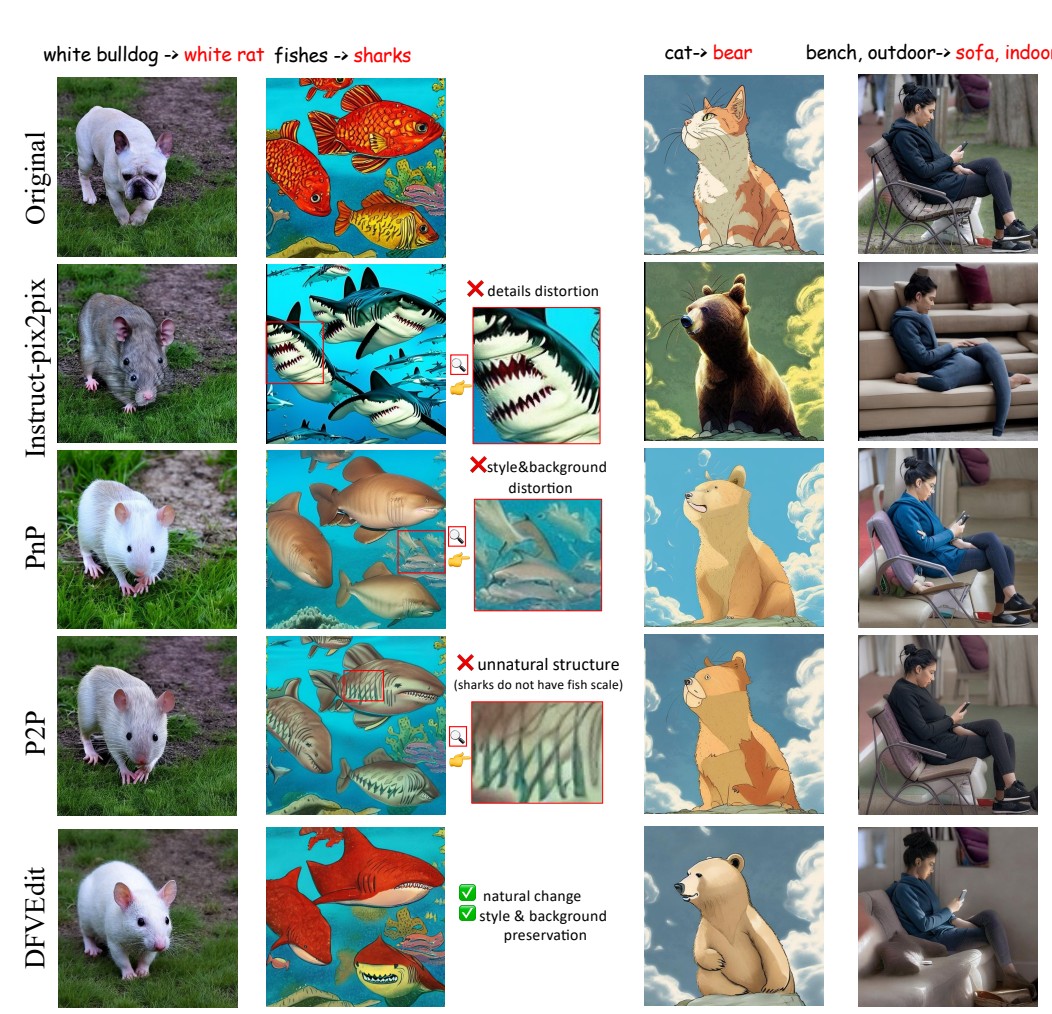

Figure F7: **Qualitative comparison results on image editing with PIE-bench.**

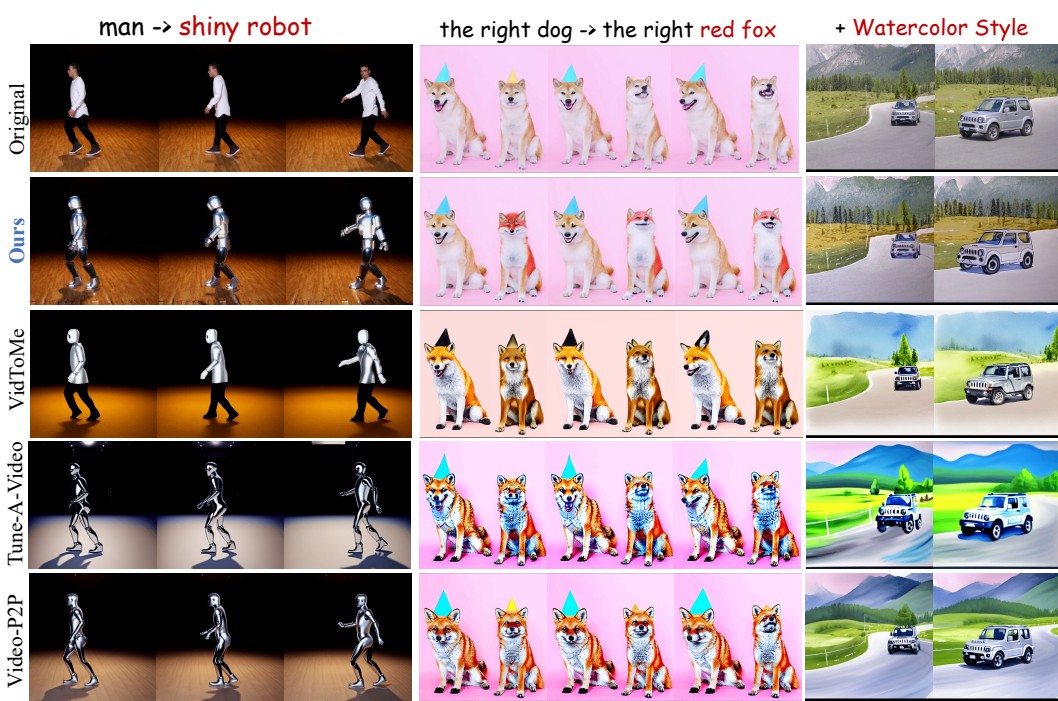

Figure F8: **More comparison results.**

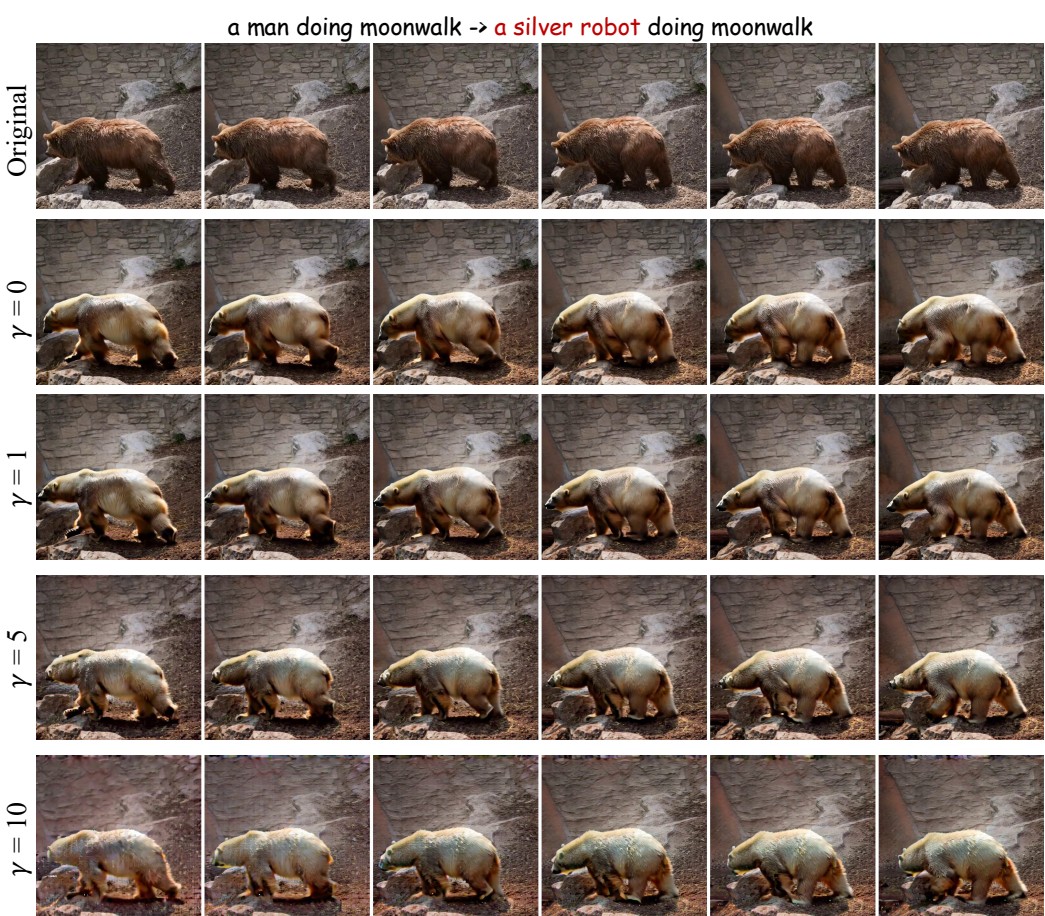

Figure F9: **Ablation results of ER on shape editing.** ER strength $\gamma$ ($\gamma = 0 \rightarrow 1$ optimal, $\gamma \geq 5$ degrades).

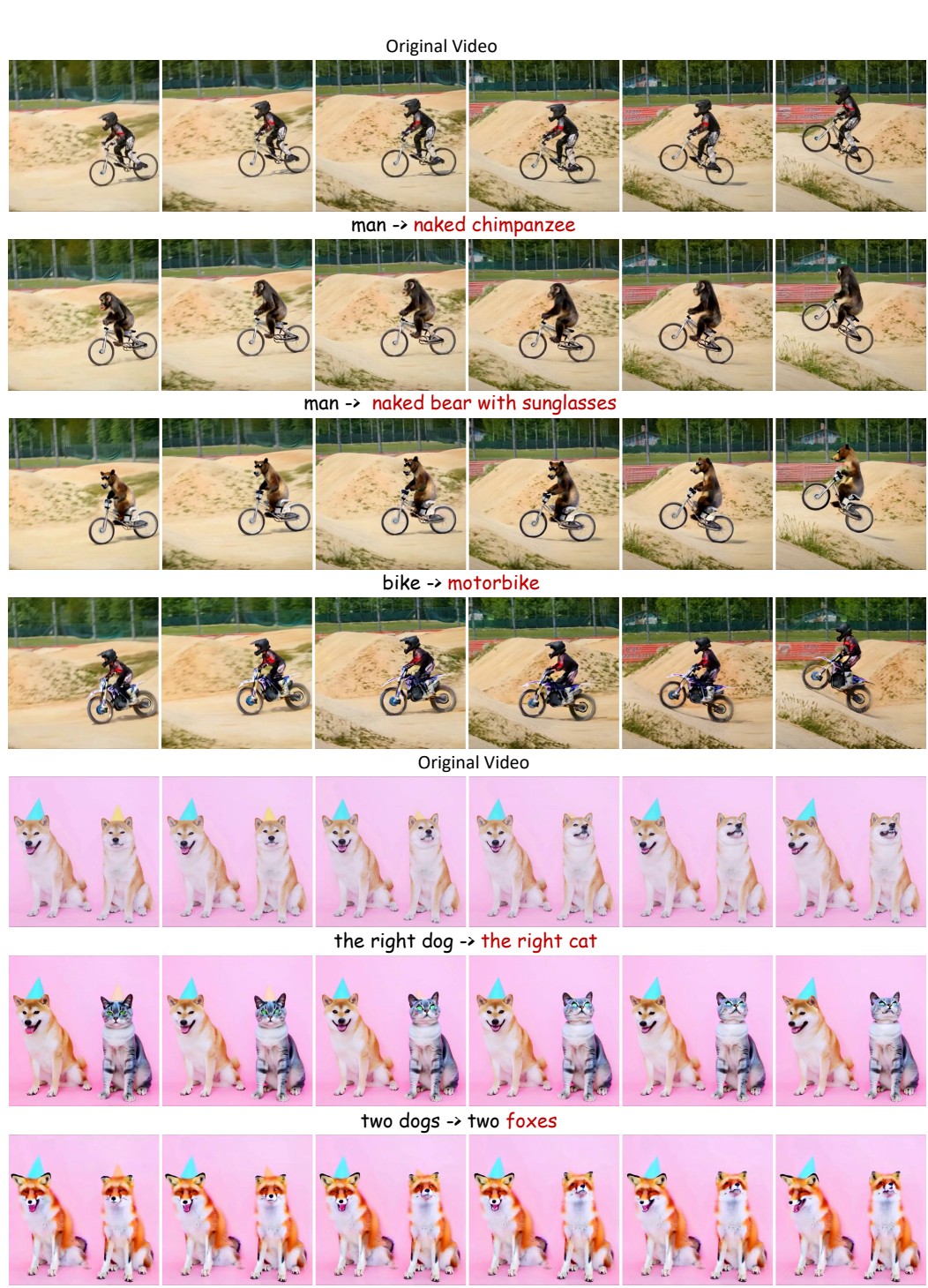

Figure F10: **Multi-object editing results.** DFVEdit performs well on multi-object editing across various scenarios: (1) complex dynamic interactions (person-vehicle) with cluttered backgrounds, and (2) fine-grained object manipulation with detailed motions (two dogs).

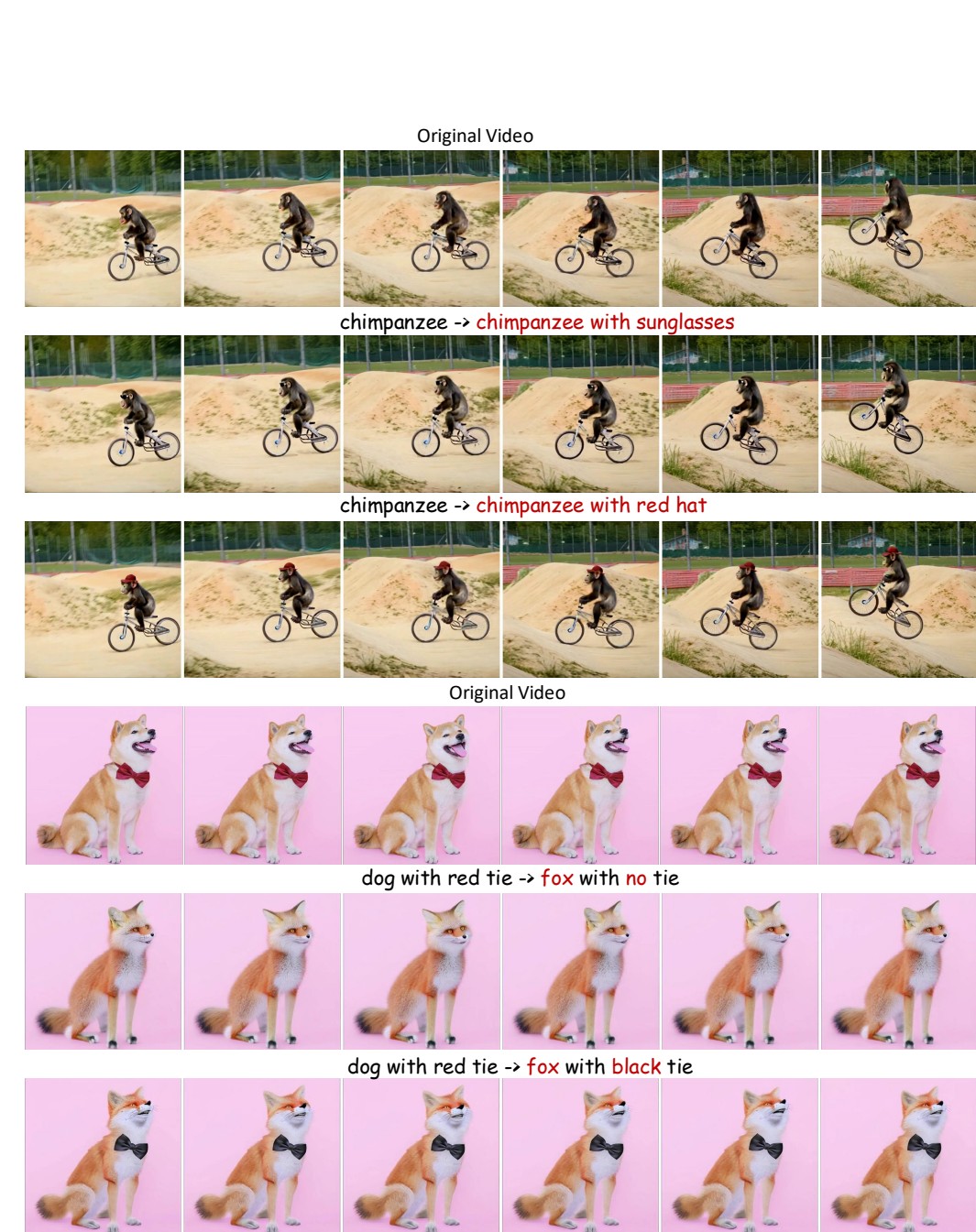

Figure F11: **Attribute editing results.**

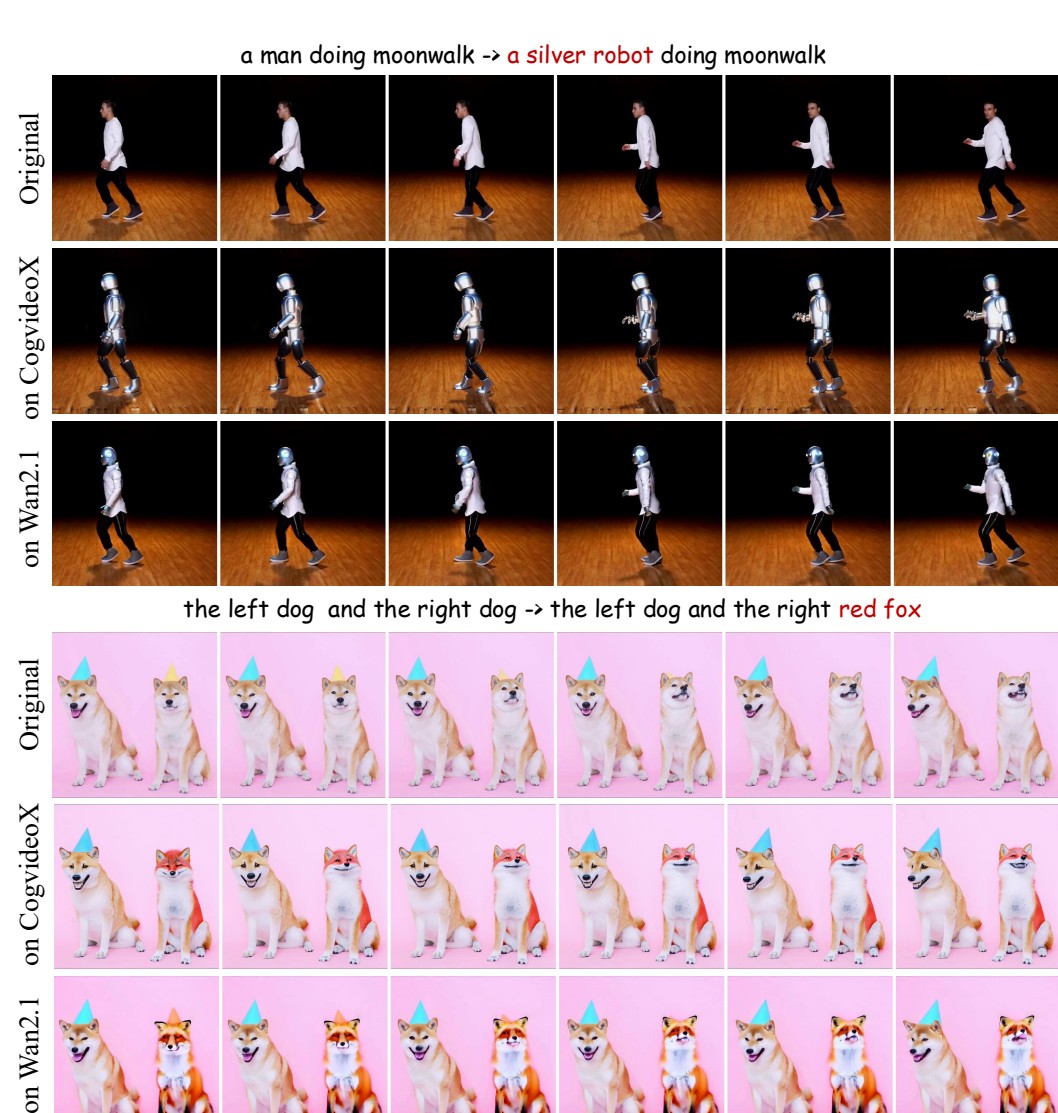

Figure F12: **More extension experiment results.**

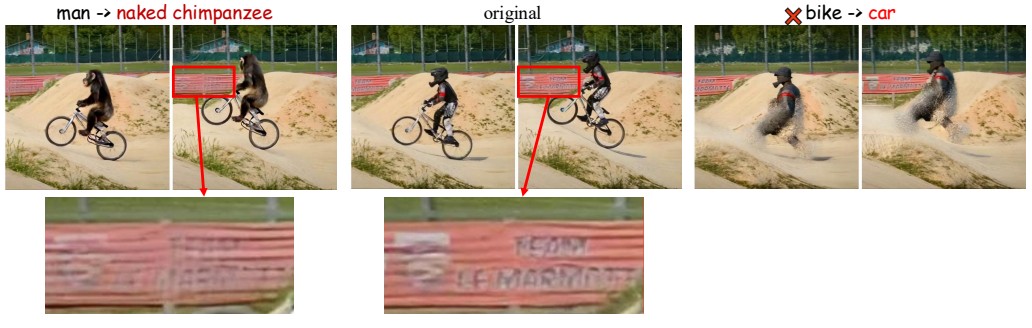

Figure F13: **Limitation.**

