# OpenReview forum: "DFVEdit: Conditional Delta Flow Vector for Zero-shot Video Editing"
_ICLR.cc/2026/Conference — Submitted to ICLR 2026_

### Official Review · Reviewer_sFW3 · 2025-10-24

**Soundness:** 2
**Presentation:** 3
**Contribution:** 2
**Rating:** 4
**Confidence:** 4

**Summary:**

This paper introduces DFVEdit, a training-free zero-shot video editing framework for Video Diffusion Transformers (Video DiTs). Unlike attention-engineering-based approaches that require high memory and computational cost, DFVEdit reformulates the editing process under a continuous flow perspective and directly manipulates latent representations instead of attention maps. The key innovation is the Conditional Delta Flow Vector (CDFV), an unbiased estimator of the latent flow difference between source and target prompts. The model further integrates Implicit Cross-Attention (ICA) and Embedding Reinforcement (ER) to enhance spatiotemporal coherence and prompt fidelity. Experiments on CogVideoX and Wan2.1 demonstrate strong editing fidelity, temporal consistency, and 20×–85% efficiency gains over previous zero-shot methods.

**Strengths:**

1. Unification of diffusion sampling and video editing under continuous flow dynamics.
2. Training-free and efficient, avoiding attention engineering with major gains in VRAM and latency.
3. CDFV formulation provides theoretical grounding for latent editing dynamics.
4. Extensive experiments on multiple base models (CogVideoX, Wan2.1) showing generalization and scalability.

**Weaknesses:**

1. Theoretical assumptions remain unverified.
The paper claims the CDFV to be an “unbiased” flow estimator, yet does not empirically validate this property or its convergence stability. It would strengthen the work to provide quantitative analysis or synthetic experiments demonstrating the unbiasedness.

2. Limited novelty beyond integration.
Although the continuous-flow formulation is elegant, it largely reformulates existing diffusion mathematics rather than proposing fundamentally new theory. The contribution lies mainly in design efficiency and practical deployment.

3. Partial dependence on implicit cross-attention.
The ICA module still indirectly relies on attention maps (albeit derived from full attention). Thus, the “attention-free” claim is somewhat overstated—memory reduction is significant but not absolute.

4. Limited evaluation scope.
The dataset is mostly limited to DAVIS2017 and short Pexels clips; long videos or open-domain scenes are not tested. It remains unclear whether DFVEdit maintains temporal stability across longer durations or higher frame rates.

5. Comparative fairness concerns.
Some baselines (e.g., FateZero, TokenFlow) were directly re-implemented on CogVideoX without architecture adaptation, which might favor DFVEdit’s efficiency comparisons. A discussion of reimplementation fairness would improve transparency.

6. Minor clarity issues.
The mathematical derivations (Eqs. 8–13) could benefit from clearer notation and better linking between theoretical constructs (e.g., Δvₜ vs ∇log P(·, t)).

**Questions:**

1. How sensitive is DFVEdit to inaccuracies in flow estimation or text embedding alignment?
2. Does ICA extraction from 3D full attention introduce any temporal artifacts when editing long sequences?
3. Could DFVEdit extend naturally to multimodal or audio-conditioned video editing?
4. What are the runtime and memory trade-offs between CogVideoX and Wan2.1 backbones?

---

### Official Review · Reviewer_7GLJ · 2025-10-28

**Soundness:** 3
**Presentation:** 2
**Contribution:** 2
**Rating:** 4
**Confidence:** 3

**Summary:**

This paper uses pre-trained diffusion or flow-matching-based video text-to-video models and introduces a new video-editing method. This is again done with a flow-based formulation, which is derived with similar methods to the original flow matching. Lots of qualitative and quantitative evaluations show the methods good performance.

**Strengths:**

Paper gives a method for using both pre-trained diffusion and fm T2V-models for video editing that is well grounded in the theory on flow matching/generative modelling.
The resulting videos look to be good-looking and are competitive with other competing methods.
Training-free scheme is great for plug-n-play and comparison of different T2V models.

Writing is, apart from the issue with the citations, mostly clear and concise and easy to follow.

**Weaknesses:**

ISSUE SAM: Unclear when, if ever, SAM masks are used
- Manuscript (Fig. 1) says that they can be used optionally
- Section C.3 says that it may be used for multi-object editing, such as in Figure F10.
- it never becomes clear, when exactly SAM masks are used. Ist just states them to be "optional"
- SAM masks are never really compared in their effectiveness to the ICA approach
- no principled experiment really evaluates their usage


ISSUE EMBEDRF: Difference between DFVEdit and DFVEdit w/o EmbedRF
- only negligibly small gains are observed for EmbedRF. Does EmbedRF really help that much? Can we be sure the results are robust? Were several runs conducted?
- together with table T4 it becomes apparent, that using EmbedRF does not do anything. The gains lie within the standard deviation for all reported metrics, except for CLIP-T
- is the only real "benefit" to make the videos look more stylized? This can probably not be shown by any metric



Major Presentation Issue:
- None of the citations (with \cite) flow properly with the text and have been used with no regard to the ICLR template. Many if not most should have been put in parantheses. This destroys the flow of reading the paper in a major way!

More Minor Presentation Issues:
  - Figure F9: the prompt mentions "a man doing moonwolk", but the example given here is the video of the bear. The entire figure seems misplaced or wrong?

  - Typos:
    - line 185/186: Wrong verb conjugation: "f(x,t) is the drift coefficient corresponds to..."
    - wrong equation linked in line 211. Eq. 17 is in the appendix. Do you mean eq. 5?
    - wrong equation linked in line 217. Eq. 31 is in the appendix. Do you mean eq. 6?
    - line 248: sentence weird?: "if we set winner process of Z_0 and \hat{Z}_0 is equal, then ...."
    - inconsistent references: some have no dates (see Cong et al. or Yang et al.)
    - line 1099: "We thank the reviewer for this suggestion". This is a first draft at ICLR. Any mentions of previous reviewing-iterations at other venues should not be in the manuscript anymore!

**Questions:**

Regarding ISSUE SAM:
1) When exactly are SAM masks used? Why was SAM not evaluated as an actual alternative masking scheme, when it is described as an important, yet optional, part of the method?

Regarding ISSUE EMBEDRF:
2) Why was this kept, when performance gains were easily within one standard deviation?

---

### Official Review · Reviewer_KCoA · 2025-10-30

**Soundness:** 3
**Presentation:** 3
**Contribution:** 3
**Rating:** 4
**Confidence:** 4

**Summary:**

This paper introduces DFVEdit, a novel zero-shot video editing method specifically tailored for modern Video Diffusion Transformers (Video DiTs). The core contribution is a flow-transformation framework that operates directly on latents, bypassing the need for computationally expensive attention modification or model finetuning. By unifying the editing and sampling processes under a continuous flow perspective, the method proposes a Conditional Delta Flow Vector (CDFV) to estimate the transformation from the source to the target video. This approach, enhanced by Implicit Cross-Attention (ICA) guidance and Embedding Reinforcement (ER), achieves state-of-the-art results in fidelity and consistency while offering speed-up and memory reduction.

**Strengths:**

-The primary strength lies in its conceptual shift away from attention engineering. Instead of manipulating the internal query/key/value matrices, the method cleverly reframes editing as a continuous flow transformation in the latent space. Conditional Delta Flow Vector (CDFV) as a theoretically-backed estimate of the "delta" between the source and target latents is an clever thing that directly helps the use of Video DiTs.

- The paper is well-written and clearly structured. The motivation for the work is well established  in the first figure.

- The authors provide strong motivation for their work, focusing on the critical need for an efficient editing solution for Video DiTs. The evaluation is thorough, using a well-chosen suite of metrics to measure distinct properties: CLIP-F for temporal consistency, E_warp for motion fidelity, M.PSNR and LPIPS for fidelity and background preservation, and CLIP-T for prompt alignment. This comprehensive quantitative and qualitative analysis, along with a user study, makes a convincing case for the method's superiority.

**Weaknesses:**

- The reported CLIP-F score of 0.9924 (Table 1) is exceptionally high. While this is presented as a strength, a score this close to 1.0 could imply that the edited frames are almost identical to each other, suggesting the CLIP-F metric may not be sensitive enough to detect subtle, fine-grained changes or potential flickering. It seems unlikely for a video to be meaningfully edited and still retain this level of inter-frame similarity unless the video was already static. It would be beneficial for the authors to provide a more detailed, one-by-one results breakdown for the evaluation dataset, or at least discuss this high score and why it doesn't indicate a lack of meaningful editing.

- The comparison setup could be strengthened. The paper's method (DFVEdit) is applied to Video DiT backbones (CogVideoX-5B, Wan2.1-14B). However, many of the baselines (e.g., FateZero, TokenFlow, VideoDirector) are evaluated on their original, often U-Net-based backbones like Stable Diffusion 1.5 (as noted in Appendix D.5 and Table T2). While the paper does test extending some baselines to CogVideoX (Fig 1b) to show they are computationally infeasible, the primary qualitative and quantitative comparisons are between methods running on different underlying generative models. Text-to-Video (T2V) models or methods specifically designed for Video DiTs would serve as a more direct and fair comparison group to truly isolate the contribution of the editing method (DFVEdit) versus the power of the backbone model (Video DiTs).

- The paper does not clearly state the total size of the evaluation dataset used for the main quantitative results in Table 1. Appendix D.1 mentions using the public DAVIS 2017 dataset and Pexels videos. Appendix D.2 mentions "10 DAVIS videos, 30 diverse prompts" for the M.PSNR metric, and Appendix D.3 mentions "80 video-prompt pairs" for the user study. However, the total number of videos used to calculate CLIP-F, E_warp, LPIPS, and CLIP-T in Table 1 is not specified. Clarifying the scale of the automated evaluation would help in assessing the robustness of the reported quantitative claims.

**Questions:**

Please see weaknesses

---

### Official Review · Reviewer_eh2p · 2025-11-03

**Soundness:** 2
**Presentation:** 2
**Contribution:** 2
**Rating:** 4
**Confidence:** 3

**Summary:**

The authors provide DFVEdit for zero-shot video editing method for Video Diffusion Transformers that bypasses the computational overhead of attention modification and fine-tuning. It operates directly on latents by unifying editing and sampling through a continuous flow perspective and using the Conditional Delta Flow Vector (CDFV). The authors additionally provide Implicit Cross-Attention Guidance for masking latents and Target Embedding Reinforcement to amplify editing text embeddings.

**Strengths:**

- The method is model-agnostic, model-training free and significantly outperforms existing methods in terms of computational efficiency.
- The authors conduct comprehensive comparisons and ablation studies.

**Weaknesses:**

- Please use the correct citation format (e.g., \cite, \citep).
- Can the authors provide pseudo code or a simplified algorithm section  for CONDITIONAL DELTA FLOW VECTOR?
- Please clarify which parts of the 3.1 UNIFIED CONTINUOUS FLOW PERSPECTIVE ON SAMPLING AND EDITING are the contribution of the paper. For example, L192-194 are already widely known to the community (e.g., Diffusion Meets Flow Matching: Two Sides of the Same Coin, 2024).
- Are the baseline methods employing all the same backbone model? It's not surprising that T2V methods are outperforming T2I-based video edting methods.
- Except for SDEdit baseline, it appears only the proposed method is deployed on CogVideoX and Wan models, which are far better models than the baseline methods use. Can this be evaluated as a fair comparison?
- The details on base model should appear in the main paper, no in supplementary sections.
- If the method binarizes cross attention section from full-attention map, and applies masking to prevent editing unintended region, how does this enable flexible editing of global attribute or style edting?
- In L299-300, the authors state 'We observe that in 3D Full-Attention, the effect of text embeddings diminishes as frame length increases.', can the authors show this phenomenon both qualitatively and quantitatively? Isn't it attributed to deviating from training distribution for the number of frames? I am quite not convinced the extensive number of frames actually have correlation with the effect of text embeddings. Can the authors provide theoretical ground for this?
- Can the authors provide similar re-weighting methods for image/video editing that have similar philosphy as the 'Target Embedding Reinforcement'.

**Questions:**

Please address the weakness.

---

### Meta-Review · Area_Chair_TVhq · 2026-01-06

**Summary:**

No rebuttal for the reviewers' comments. Most of the reviewers are negative. In particular, the clarify of the presentation, the baseline comparison, and the writing were some examples of the concerns.

**Reviewer Concerns:**

.

**Reviewer Scores:**

.

---

### Decision · Program_Chairs · 2026-01-26

Reject